# Structural basis for the modulation of MRP2 activity by phosphorylation and drugs

Tiziano Mazza [1,2,3], Theodoros I. Roumeliotis [4,9], Elena Garitta [5,9], David Drew[6], S. Tamir Rashid[7], Cesare Indiveri [3,8], Jyoti S. Choudhary [4], Kenneth J. Linton [5] & Konstantinos Beis [1,2] ✉

Multidrug resistance-associated protein 2 (MRP2/ABCC2) is a polyspecific efflux transporter of organic anions expressed in hepatocyte canalicular membranes. MRP2 dysfunction, in Dubin-Johnson syndrome or by off-target inhibition, for example by the uricosuric drug probenecid, elevates circulating bilirubin glucuronide and is a cause of jaundice. Here, we determine the cryo-EM structure of rat Mrp2 (rMrp2) in an autoinhibited state and in complex with probenecid. The autoinhibited state exhibits an unusual conformation for this class of transporter in which the regulatory domain is folded within the transmembrane domain cavity. In vitro phosphorylation, mass spectrometry and transport assays show that phosphorylation of the regulatory domain relieves this autoinhibition and enhances rMrp2 transport activity. The in vitro data is confirmed in human hepatocyte-like cells, in which inhibition of endogenous kinases also reduces human MRP2 transport activity. The drug-bound state reveals two probenecid binding sites that suggest a dynamic interplay with autoinhibition. Mapping of the Dubin-Johnson mutations onto the rodent structure indicates that many may interfere with the transition between conformational states.

MRP2 (multidrug resistance-associated protein, 2) is a primary active transporter of the ATP-binding cassette (ABC) class[1]. There are nine C-subfamily members in humans with distinct physiological roles ranging from a chloride channel (the cystic fibrosis transmembrane regulator (CFTR); ABCB7) and a regulator of potassium channels (the sulphonyl urea receptors 1 and 2; ABCC8 and 9) to exporters of organic anions (MRP1-5).

MRP2 is localised to the apical membranes of polarised cells such as hepatocytes, enterocytes, pneumocytes and proximal tubule cells of the kidney. Loss of MRP2 function manisfests primarily as a liver disease caused by the failure to export bilirubin glucuronide, the hepatic

product of heme breakdown. Failure to eliminate the compound across the canalicular membrane into the bile ultimately results in jaundice, the primary feature of Dubin–Johnson syndrome, after the conjugated bilirubin is exported back into the circulation by the basolateral MRP3[2]. Over 40 mutations have been identified in the *ABCC2* gene, some of which are neutral, others are non-sense or mis-sense mutations causing the absence of the MRP2 protein from the hepatocyte apical membrane or MRP2 with compromised transport activity, respectively[3,4].

Other transport substrates of MRP2 have been identified in vitro and include glutathione- sulfate-, and glucuronide- conjugated

[1]Department of Life Sciences, Imperial College London, SW7 2AZ London, UK. [2]Rutherford Appleton Laboratory, Research Complex at Harwell, Didcot, Oxfordshire OX11 0FA, UK. [3]Department DiBEST (Biologia, Ecologia, Scienze Della Terra) Unit of Biochemistry and Molecular Biotechnology, University of Calabria, 87036 Arcavacata di Rende, Italy. [4]Functional Proteomics group, Chester Beatty Laboratories, The Institute of Cancer Research, London SW3 6JB, UK. [5]Blizard Institute, Faculty of Medicine and Dentistry, Queen Mary University of London, E1 2A London, UK. [6]Department of Biochemistry and Biophysics, Stockholm University, 10691 Stockholm, Sweden. [7]Department of Metabolism, Digestion & Reproduction, Imperial College London, W12 0NN London, UK. [8]CNR Institute of Biomembranes, Bioenergetics and Molecular Biotechnology (IBIOM), 70126 Bari, Italy. [9]These authors contributed equally: Theodoros I. Roumeliotis, Elena Garitta. ✉e-mail: kbeis@imperial.ac.uk

metabolites of Phase II metabolism. These include conjugated leukotriene C4 (conjugated LTC4) and therapeutic anticancer drugs such as paclitaxel and cisplatin[5–11]. The physiological relevance of drug transport remains unclear but animal studies have shown that engineered expression of MRP2 in implanted cancer cells confers resistance to cisplatin[12] and in a small clinical study MRP2 expression levels in hepatocellular cancer patients correlated with a reduction in cisplatin-induced tumour necrosis suggesting that MRP2 is a potential determinant of multidrug resistance[13].

The activity of several ABCC transporters are known to be mediated by protein kinases. For example, phosphorylation of MRP1 by the protein kinase CK2 alpha (CK2α) leads to increased efflux of doxorubicin[14], while CFTR is activated by protein kinases C and A to allow ATP-dependent channel opening[15,16]. In addition to phosphorylation, transport activity can also be modulated by drug-drug interactions. For MRP2 the uricosuric agent probenecid and the diuretic furosemide have been shown to inhibit the efflux of N-ethylmaleimide glutathione in vitro[17]. Jaundice is also a reported off-target side effect of probenecid[18].

In this work, we screen for mammalian homologues of human MRP2 and we identify the *Rattus norvegicus* Mrp2 (rMrp2) as suitable for structural studies (80% sequence identity to its human ortholog). We determine the cryo-EM structure of rMrp2 in a nucleotide-free state (3.21 Å resolution) and a drug-bound state (3.45 Å resolution) with the uricosuric agent probenecid. The nucleotide-free state is in an auto-inhibited state. Functional data coupled to mass spectrometry-based proteomics identify phosphorylation sites that likely modulate the activation of the protein and transport data with rMrp2 reconstituted in liposomes show a strong correlation between the phosphorylation state of the transporters and transport activity. This observation is corroborated in a cellular system using human hepatocytes derived from induced-pluripotent stem cells (iPSCs) which shows a reduction in MRP2 transport activity following inhibition of protein kinase activity. The probenecid bound structure reveals two drug binding sites within the rMrp2 cavity. The mutations that cause Dubin–Johnson syndrome are mapped onto the rMrp2 structure, which provides insights on how they contribute to MRP2 dysfunction by interfering with conformational changes along the transport cycle rather than substrate binding. Based on this work, we propose a mechanism for the modulation of the MRP2 activity by phosphorylation and drugs.

## Results

### cryo-EM structure of nucleotide-free rMrp2

We expressed rMrp2 in *Saccharomyces cerevisiae* as part of our screen to identify homologues suitable for structural studies[19]. Extraction of rMrp2 with Lauryl Maltose Neopentyl Glycol (LMNG) and cholesteryl hemisuccinate (CHS) and further purification into glyco-diosgenin (GDN) resulted in highly pure and monodisperse sample suitable for functional and structural studies (Fig. 1). The recombinant rMrp2 was reconstituted in destabilised liposomes consisting of bovine liver lipid extract, and it displayed a basal ATPase activity of 12.43 ± 3 nmol/mg/min that could be stimulated by the clinical drugs probenecid and methotrexate by 2.9 and 1.5 fold, respectively[17] (Fig. 1). Single-particle cryo-EM analysis of the nucleotide-free rMRP2 resulted in a three-dimensional reconstruction at an overall resolution of 3.21 Å (Fourier shell correlation (FSC) = 0.143 criterion; Supplementary Fig. 1 and Supplementary Table 1). The maps showed density with good connectivity and the presence of side chains that could be easily interpreted for model building. The rMrp2 (AF-Q63120-F1) AlphaFold2 model was used as a starting model. We have built an atomic model for rMrp2 that consists of a small N-terminal transmembrane domain (TMD0) that is linked to two transmembrane domains (TMD), TMD1 and TMD2, by the lasso linker ($L_0$). TMD0 is composed of a bundle of 5 transmembrane helices (TMs); the interface of TMD0 and TMD1 is occupied by several CHS molecules (not modelled as only part of the

sterol moiety was resolved) (Fig. 1). The $L_0$ linker has been shown to facilitate correct folding and trafficking of ABCC transporters[20,21]. Each TMD consists of 6 TM helices, enclosing the pocket for substrates and drugs binding, and they are linked to two nucleotide binding domains (NBDs), NBD1 and NBD2, where ATP binding and hydrolysis provides the free energy required for the transport cycle completion. NBD1 and NBD2 form an interface with the TMD1 and TMD2 via the intracellular loop 4 (ICL4) and 2 (ICL2), respectively; ICL2 and ICL4 are known as coupling helices and their role is to transmit conformational changes to the TMDs, associated with either NBD dimerisation or disengagement upon ATP binding or hydrolysis, respectively[22]. In the absence of nucleotides, rMrp2 adopts an inward-facing conformation with the TMDs open to the cytosol where substrates and drugs can access it. An unusual feature of the rMrp2 cryo-EM maps was the presence of helical-like density within the TMD1 and TMD2 interface that corresponds to the regulatory domain (R-domain) (Fig. 2); MRP2 and other members of the ABCC family contain an R-domain that regulates their transport activity upon phosphorylation by specific kinases (CKII, PKC, PKA and PLK kinases)[23]. The R-domain is predicted to be partly helical from the AlphaFold2 model and it links NBD1 to the elbow helix (short helix at the amine-terminal of TM12 and it sits perpendicular to the TMD2), but in the predicted model it was placed at the interface of the NBDs and outside the TMD whereas in our structure it sits deep inside the TMD. The R-domain consists of residues Gly863-Thr958. Although, continuous density from NBD1 to the helical density can be observed, only the helical part, residues Ala895-Lys924, was modelled as the rest of the density is weak and without defined side chains; in addition, there is no density that links the C-terminus of the R-domain to the elbow helix. The R-domain is buried within the TMDs at the interface of TMs 8, 9, 11 and 17 and it is stabilised by several hydrogen bonds and Van der Waals interactions (Fig. 2).

The overall architecture of rMrp2 resembles that of another member of the ABCC family, the bovine Mrp1 (bMrp1)[24]. MRP1 is the closest homologue of MRP2 (49% identity between the human sequences) but it is localised in the basolateral membrane of polarised cells in most tissues with relatively low levels in liver. Although MRP1 and MRP2 have some common substrates, there are significant differences on clinical drug specificity[25]. Despite their overall fold being very similar, rMrp2 and bMrp1 can be superimposed with a root-mean-square deviation (rmsd) of 5.6 Å over 1400 Cα atoms as a result of a 26° tilt of the TMD0 away from TMD1 (Supplementary Fig. 2); excluding the TMD0 from the alignment, the two structures can be superimposed with an rmsd of 3 Å. The main conformational difference is in the NBDs, that adopt a more closed conformation in rMrp2 due to a 2.5 Å displacement of the NBD1 in tandem with ICL4 towards NBD2 (Supplementary Fig. 2). Two other members of the ABCC family that have their structures resolved are the human cystic fibrosis transmembrane conductance regulator (CFTR)[26] and the *S. cerevisiae* Ycf1 transporter[27,28], but these are functionally distinct from rMrp2 functioning as a chloride channel and cadmium metal transporter, respectively. The most striking difference between rMrp2, CFTR and Ycf1 is the conformation of the R-domain; the R-domain in bMrp1 was not resolved. In CFTR, the dephosphorylated R-domain adopts a helical structure similar to rMrp2 (Fig. 2); in the CFTR structure, a 19 residue helix, presumed to be from the R domain, is resolved and it is localised in the cytosolic face of the TMDs making contact with the intracellular regions of TMs 14 and 15. The CFTR is in an inactive-like conformation as the R-domain sits between NBD1 and NBD2 and prevents them from forming an interface[26]. In phosphorylated Ycf1, 30 residues of the R-domain were resolved, it adopts an extended conformation outside the interface of TMD1/2 where it is stabilised by residues on the surface of NBD1[27,28]. In the phosphorylated CFTR the R-domain retains its helical structure and it is stabilised by interactions with residues from TMs 14, 15 and 17[29] (Fig. 2). As the R-domain of rMrp2 is deep inside the TMD1/2 interface, it points to an inactive conformation that would

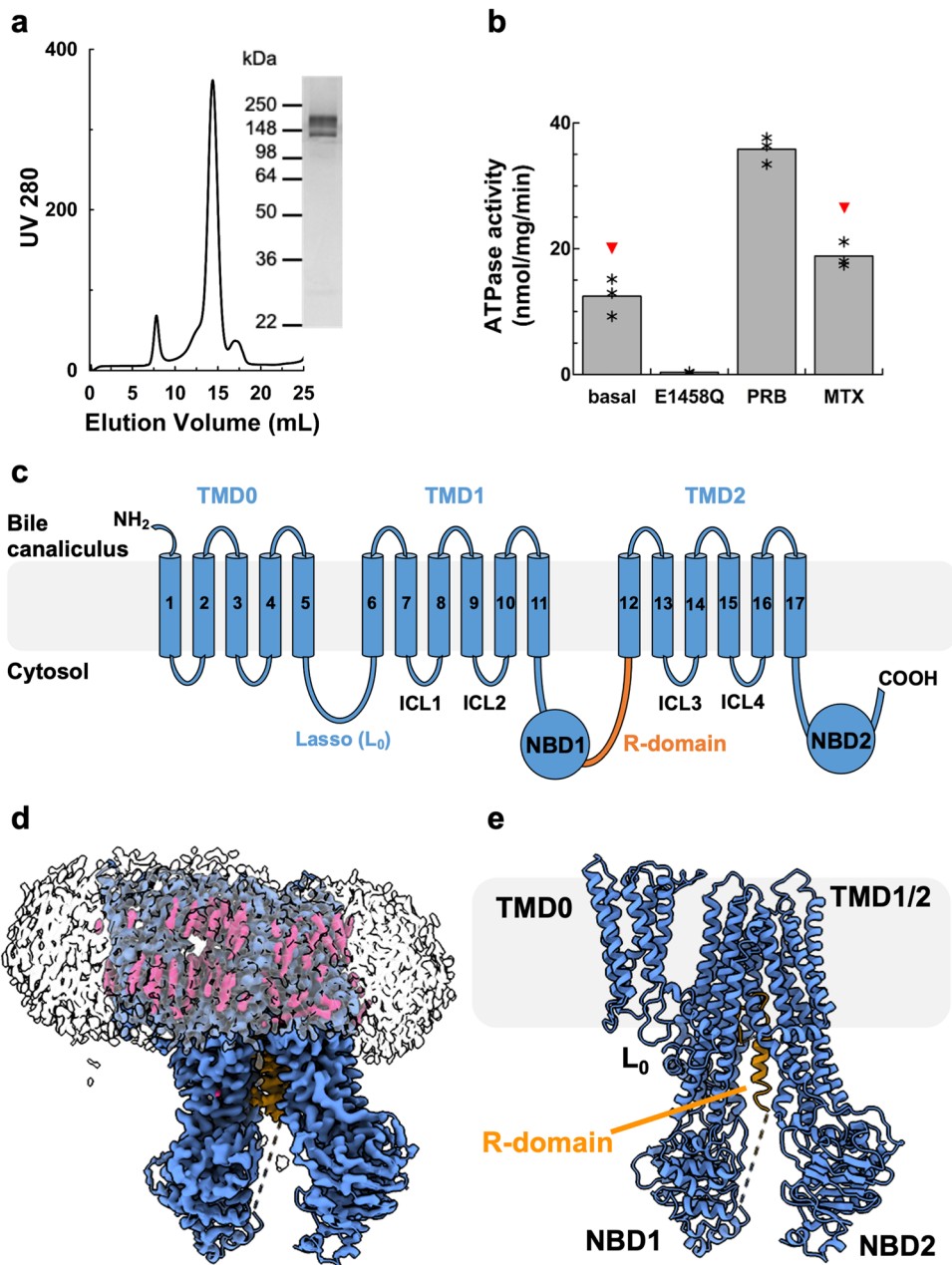

**Fig. 1 | Cryo-EM structure of nucleotide-free rMrp2. a** Chromatogram and SDS-PAGE analysis of Size Exclusion Chromatography of purified rMrp2. Representatives from three purifications. **b** Effect of drugs on the basal ATPase activity of rMrp2 reconstituted in destabilised liposomes. The basal ATPase activity of rMrp2 can be stimulated by 1 mM probenecid (PRB) and 0.1 mM methotrexate (MTX). The E1458Q mutant displays no ATPase activity. Results are represented as means with individual data points indicated by asterisc, from three independent experiments. Significantly different as estimated by one way ANOVA ($p = 2.32 \times 10^{-7}$) followed by Tukey test $p < 0.001$ (red triangle: $p < 0.01$)). **c** MRP2 topology with the key domains highlighted. **d** Ab initio cryo-EM map of rMrp2 at 3.21 Å resolution. Continuous density can be observed for the different rMrp2 domains. The CHS molecules and GDN micelle are shown in pink and transparent grey, respectively. **e** The nucleotide-free rMrp2 atomic model displays an inward-facing conformation with the TMD cavity occupied by the R-domain; the helical R-domain is shown in orange. Source data are provided as a Source Data file.

prevent bilirubin glucuronide or drug binding and the formation of an NBD:NBD interface. This is similar to the situation with CFTR where the weak density that links NBD1 and the helical portion of the R-domain is found close to NBD1 that would prevent it from interacting with NBD2 upon ATP binding, due to steric clashes.

### Regulation by phosphorylation
Considering the conformation of the R-domain in rMrp2, we investigated the phosphorylation state of the purified protein and probed how phosphorylation regulates its activity. The analysis of the purified rMrp2 by SDS-PAGE phosphostaining showed that the recombinant

protein is phosphorylated after purification (Fig. 3). Mass spectrometry-based proteomics analysis of the purified rMrp2 provided more quantitative data; it showed that rMrp2 is phosphorylated at several predicted sites (Fig. 3, Supplementary Fig. 3 and Source Data file), including residues of the R-domain, Thr869, Ser874 and Thr886, with the exception of Ser922 (only a very weak ambiguous signal could be detected) and Ser926 (Fig. 3); the partially phosphorylated protein will be referred as rMrp2R-*. The R-domain contains several phosphorylation motifs for the CKII, PKC and PLK kinases[30]. Although, similar kinases are present in *S. cerevisiae*, it is very likely that subtle changes in the sequence or conformation of rMrp2 may prevent

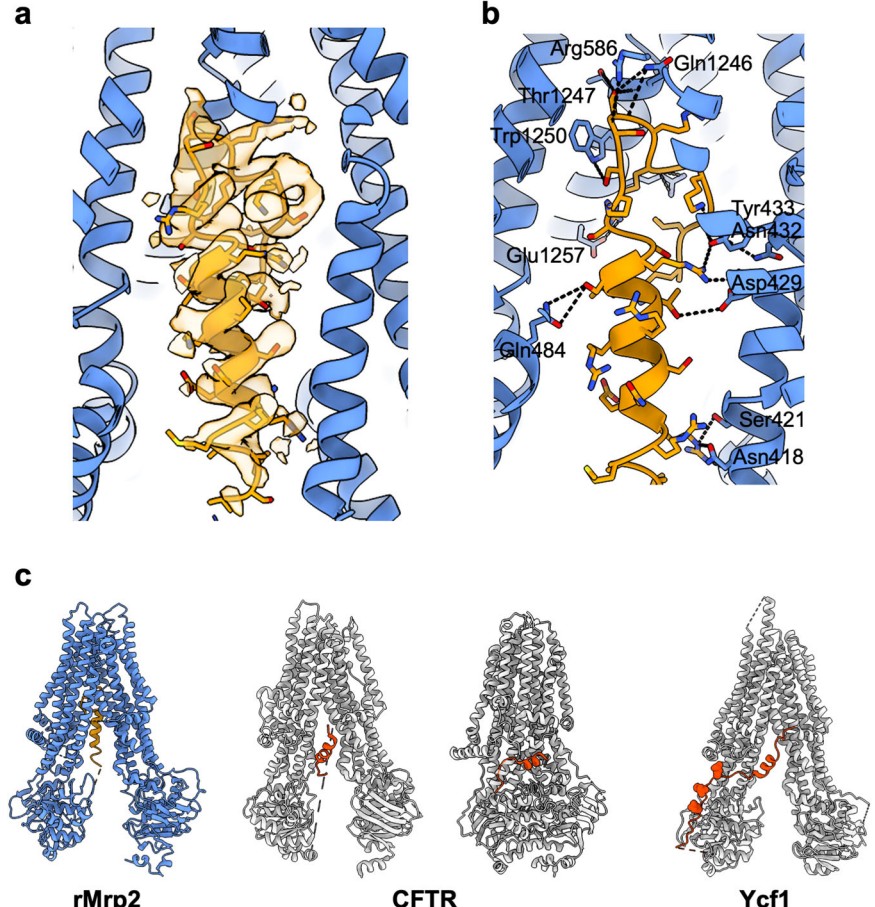

**Fig. 2 | rMrp2 exhibits an autoinhibited conformation. a** Continuous helical density is observed for the R-domain (shown in orange cartoon for reference). **b** The R-domain is stabilised by several hydrogen bonds and van der Waals interactions within the TMD. The side chains involved in hydrogen bonds (shown in black dashed lines) are shown in stick. **c** Comparison of the conformation of the R-domain between rMrp2 (partially phosphorylated; (R-domain in orange)), CFTR (dephosphorylated (left panel; PDBID: 5UAK) and phosphorylated (right panel; PDBID: 6MSM); R-domain in red) and Ycf1 (phosphorylated (PDBID: 7M69); R-domain and resolved phosphosites (spheres) in red).

phosphorylation of all the predicted sites. In our cryo-EM maps, we did not resolve any phospho modified residues.

We manipulated the phosphorylation state of rMrp2 in vitro, to determine the impact of phosphorylation on its activity. Firstly, we fully dephosphorylated Mrp2$_{R-*}$ with λ-protein phosphatase (referred as rMrp2$_{R-de}$); dephosphorylation did not alter the ATPase activity of rMrp2 in destabilised liposomes (Fig. 3). Since *S. cerevisiae* is not capable of fully phosphorylating the R-domain of rMrp2, we devised a protocol for the in vitro phosphorylation of rMrp2$_{R-*}$ during purification. We incubated rMrp2$_{R-*}$ with a cell extract from human embryonic kidney 293 (HEK293) cells that contains the endogenous kinases, in the presence of phosphatase inhibitors to prevent dephosphorylation by endogenous phosphatases. The rMrp2 was further purified to remove the cell extract proteins. Phosphostaining showed a 48% enhancement of rMrp2 phosphorylation compared to rMrp2$_{R-*}$ using this in vitro protocol (Fig. 3). Quantitative analysis by mass spectrometry confirmed that in the presence of the extract, rMrp2 is fully phosphorylated including Ser922 and Ser926 in the R-domain in addition to other sites throughout the transporter (referred as rMrp2$_{R-pho}$) (Fig. 3). The basal ATPase activity of rMrp2$_{R-pho}$ showed 70% activity enhancement relative to the rMrp2$_{R-*}$ and rMrp2$_{R-de}$ and inclusion of probenecid in the assays did not stimulate the ATPase activity any further suggesting a critical role for Ser922 and Ser926 on modulating the activity of rMrp2.

Since the different levels of rMrp2 phosphorylation impacted the ATPase activity, we also investigated their influence in its transport activity. rMrp2 was reconstituted in proteolisomes and we measured the rMrp2-dependent uptake of the fluorescent substrate 5(6)-Carboxy-2′,7′-dichlorofluorescein (CDF) inside the liposomes. The Mrp2$_{R-*}$ and rMrp2$_{R-de}$ transported the CDF with a rate of 0.5 nmol/mig/30 min whereas the rMrp2$_{R-pho}$ displayed a significant increase in its transport rate of 3 nmol/mig/30 min. The increased basal ATPase activity and uptake of the CDF by the fully phosphorylated rMrp2 suggested that full phosphorylation of the R-domain relieves its inhibitory effect and prevents it from occupying the TMD1/2 interface; a similar effect was observed for the fully phosphorylated CFTR whose channel opening and ATPase activity were enhanced upon phosphorylation[26].

To further assess the effect of phosphorylation on the activity of MRP2 in a cellular context, we performed transport assays in polarised hepatocyte-like cells derived from human induced pluripotent stem cells (iPSCs) (induced hepatocytes (iHEPs)) expressing human MRP2 in the absence and presence of the broad-spectrum kinase inhibitor staurosporine, which should prevent human MRP2 phosphorylation by the endogenous kinases. The iHEPs are polarised, make and secrete bile into canaliculi formed between adjacent cells, thus mimicking the native cellular context of human MRP2. Human MRP2 can transport the CDF into the bile canaliculi that is reduced by nearly 50% in the presence of staurosporine (Fig. 4) further inferring the role of phosphorylation in modulating the activity of MRP2 (treatment of the iHEPs with staurosporine does not alter the localisation or expression levels of human MRP2 (Supplementary Fig. 4). Our transport data in proteoliposomes and iHEPs provide strong evidence how phosphorylation of the R-domain impacts substrate transport.

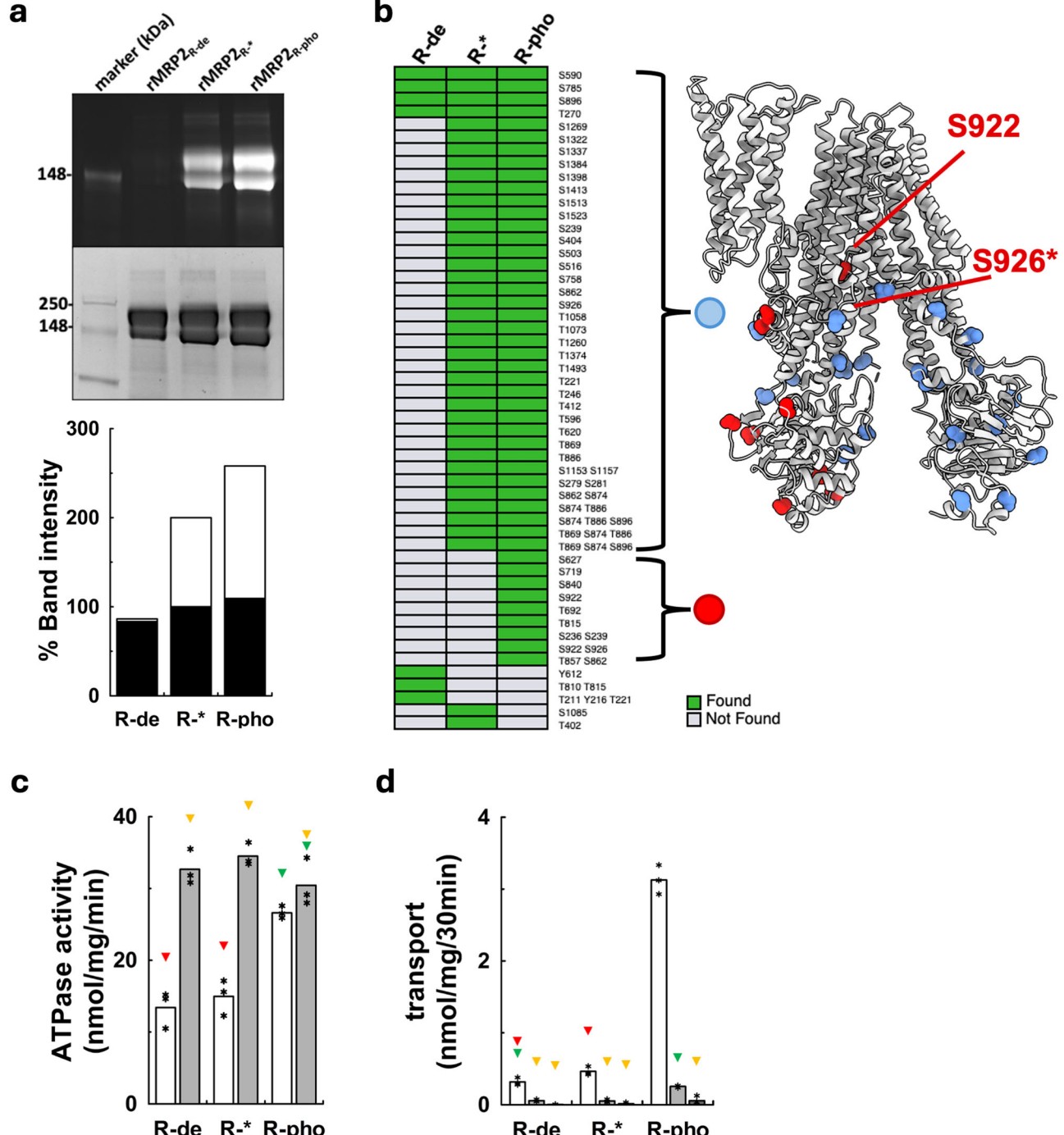

**Fig. 3 | Effect of phosphorylation on the activity of rMrp2. a** The phosphorylation state of rMrp2 was analysed by Pro-Q Diamond Phosphoprotein Stain (top panel) and Coomassie blue (bottom panel) of the SDS-PAGE. Densitometry analysis shows a 48% enhancement of phosphorylation upon in vitro treatment with the HEK293 kinase extract; white and black bars correspond to the top and bottom panels of the SDS-PAGE, respectively. Phosphorylation state abbreviations: rMrp2$_{R-*}$ (partially phosphorylated), rMrp2$_{R-de}$ (fully dephosphorylated) and rMrp2$_{R-pho}$ (fully phosphorylated). Representative SDS-PAGE from three repeats. **b** (left panel) Mass spectrometry-based phosphorylation mapping of rMrp2 protein showing presence or absence of detection across different conditions. The phosphorylated residues have been mapped onto the rMrp2 structure (right panel). The phosphorylated residues that are present in both the rMrp2$_{R-*}$ and rMRP2$_{R-pho}$ samples are shown as blue spheres. The sites that are unique to the rMrp2$_{R-pho}$ sample are shown as red spheres; the additionally phosphorylated R-domain residues are labelled for clarity (a star indicates a phosphorylated R-domain residue that has not been built in the structure). **c** Effect of probenecid on the ATPase activity of rMrp2 in

different phosphorylation states. Dephosphorylation by λ-protein phosphatase does not affect the basal ATPase (white bars), whereas full phosphorylation increases the basal ATPase by 2-fold. Probenecid can stimulate the basal ATPase activity of rMrp2$_{R-*}$ and rMRP2$_{R-de}$ but it has no additional effect on rMrp2$_{R-pho}$ (grey bars). Results are represented as means from three independent experiments. Significantly different significantly different as estimated by one way ANOVA ($p = 2 \times 10^{-7}$) followed by Tukey test $p < 0.025$ (red, yellow and green triangles:not significant). **d** rMrp2-dependent uptake of CDF in proteoliposomes. Transport was initiated by adding (empty bars and light grey bars) or not (dark grey bars) ATP-MgCl$_2$, to the samples containing (light grey) or not (empty and dark grey) 1 mM probenecid. rMrp2$_{R-pho}$ displays significantly increased transport activity relative to the rMrp2$_{R-*}$ and rMrp2$_{R-de}$ proteins. Probenecid inhibits CDF transport. Results are represented as means from three independent experiments. Significantly different significantly different as estimated by one way ANOVA ($p = 1.6 \times 10^{-19}$) followed by Tukey test $p < 0.025$ (red, yellow and green triangles: not significant). Source data are provided as a Source Data file.

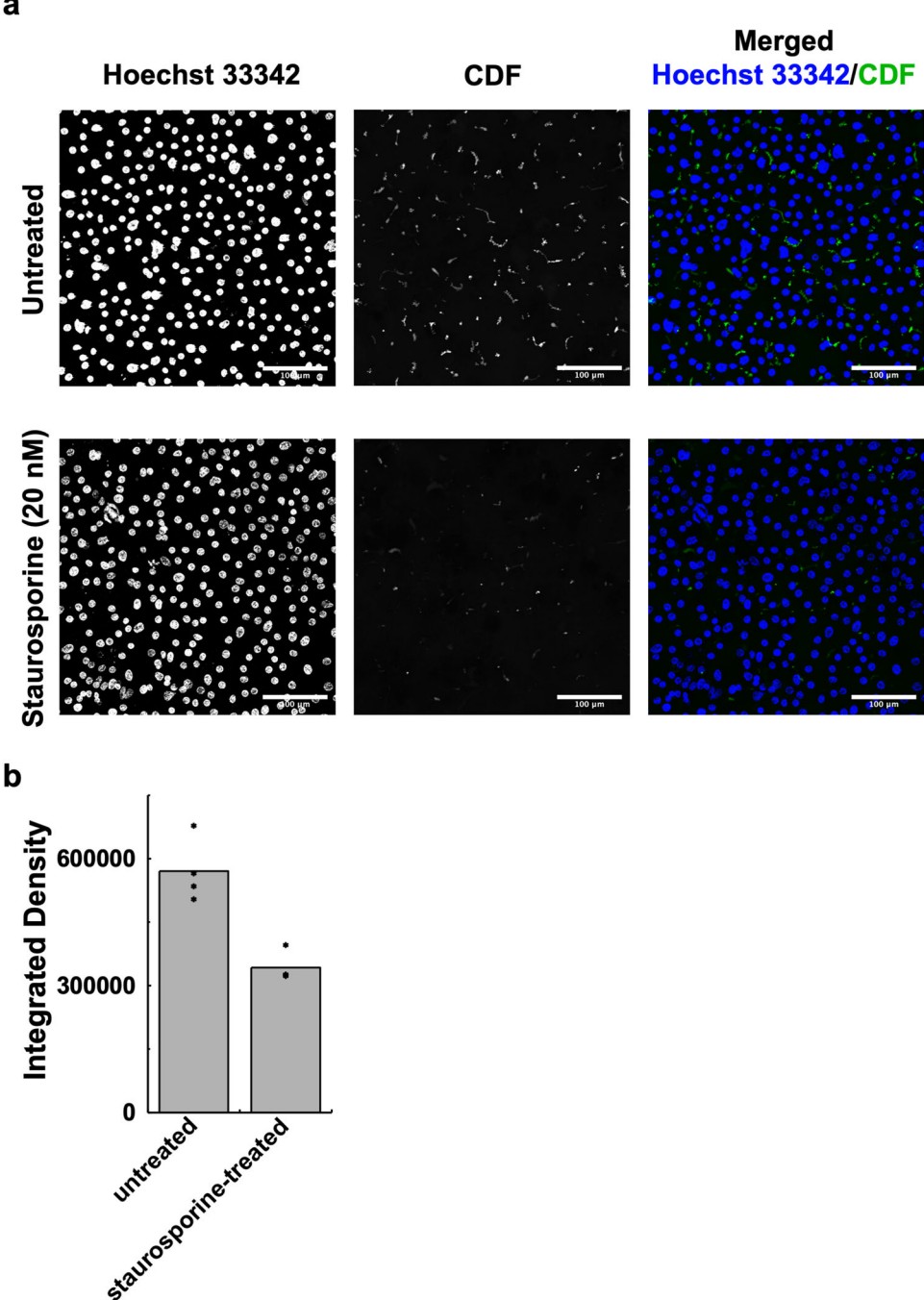

**Fig. 4 | Human MRP2 activity is reduced by kinase inhibition in hepatocyte-like cells. a** Live cell imaging of CDF (green) accumulation in the bile canaliculi of iHEPs. Nuclei are stained with Hoechst 33342 (blue). Single grayscale images and merged multicolour images are shown. **b** Integrated density image analysis shows a 50% reduction in CDF transport into the canaliculi upon treatment with staurosporine. Results are represented as means from 4 iHEP batches ($n = 3$ images per condition per batch). Significantly different as estimated by one-tail paired t-test ($p = 8.01 \times 10^{-4}$). Source data are provided as a Source Data file.

## Drug-bound rMrp2 structure

To gain an understanding of the molecular basis of drug recognition and polyspecificity by Mrp2 and its modulation by drugs, we determined its cryo-EM structure in the presence of probenecid, to 3.45 Å resolution (Fig. 5 and Supplementary Fig. 5). Probenecid is reported to act idiosynchratically on MRP2, inhibiting the transport of some substrates and stimulating the transport of others[5,17]. In our assays, probenecid stimulated the basal ATPase activity of rMrp2 but also acted as an inhibitor for the transport of CDF into liposomes that is consistent with this previously reported modulating activity (Figs. 1b and 3d). The cryo-EM map revealed clear density for two probenecid molecules and one CHS (Fig. 5). Several biochemical studies had proposed that MRP2 (and MRP1) contain two drug binding sites[31,32], but they had not been identified within the TMDs. Strikingly, the two probenecid molecules and the CHS have displaced the R-domain out of the TMD and occupy a similar space as the R-domain in our apo structure (the same rMrp2 preparation was used for both structures) (Figs. 5 and 6); no density for the R-domain can be observed in our maps suggesting a high degree of flexibility. Displacement of the R-domain from the TMD by probenecid results in a more closed overall conformation at the cytosolic half-end of the TMD and NBDs by 8 Å, respectively, whereas the otherhalf of the

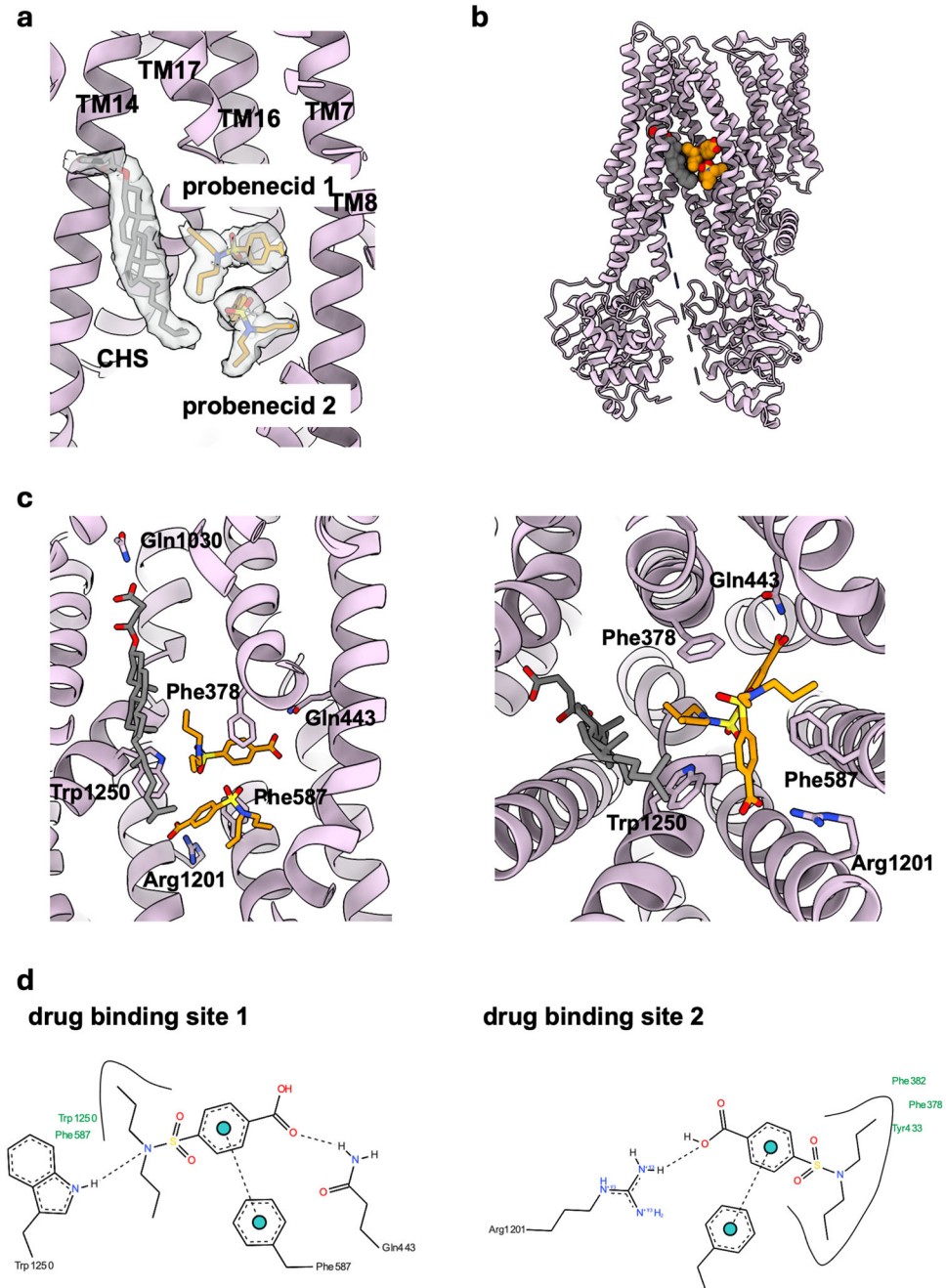

**Fig. 5 | Cryo-EM structure of drug-bound rMrp2. a** Density for two probenecid molecules (shown in orange sticks) and one CHS molecule (shown in grey sticks) within the TMD; their structures are shown for reference. **b** Binding of probenecid displaces the R-domain from the TMD. **c** The two probenecid molecules are coordinated by hydrogen bonds and π-stacking interactions within the TMD. Left panel is a view along the membrane and right panel is a view from the cytosol. **d** The probenecid bound structure defines the two drug binding sites that are in close proximity to each other.

TMD towards the membrane, where the drug density is found, displays minimal movement (Fig. 6).

The TMD region defined by TMs 7, 8, 16 and 17, form the two probenecid binding sites (Fig. 5). The probenecid in drug-binding site 1 is coordinated by a hydrogen bond between Gln443 and its carboxylate, a hydrogen bond between Trp1250 and the sulfonamide, and a π-stacking interaction between Phe378 and its benzene ring. The probenecid in site 2 is coordinated by a hydrogen bond between its carboxylate and Arg1201 and a π-stacking interaction between Phe587 and its benzene ring (Fig. 5). The CHS molecule is coordinated mostly by van der Waals interactions and a hydrogen bond between its carboxylate and Gln1030.

In the bMrp1 structure in complex with conjugated leukotriene C4 (LTC4)[24], the glutathione (GSH) moiety is coordinated by positively charged residues in the P-pocket and its lipid tail by van der Waals interactions within the H-pocket. In comparison with our rMrp2 structure, the probenecid binding site 1 overlaps with the P-pocket of bMrp1 whereas the drug binding site 2 is distal to both the drug binding site 1 and LTC4 binding site (Fig. 6); the CHS binding site overlaps with the H-pocket of bMrp1. The drug-bound rMrp2 has a very similar overall conformation to the substrate-bound bMrp1, and the two structures can be superimposed with an rmsd of 2.8 Å (excluding the TMD0), suggesting that the TMD is not required to undergo significant conformational changes to bind substrates and

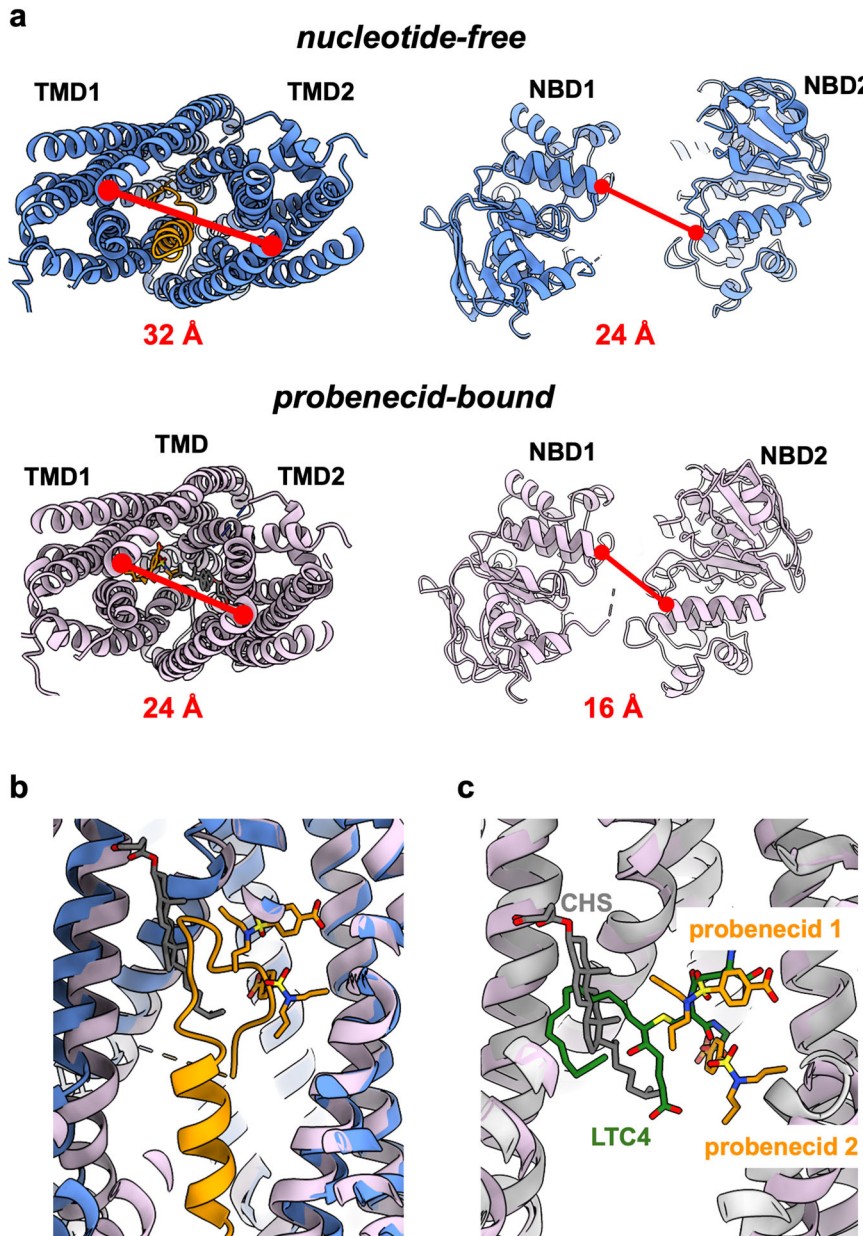

**Fig. 6 | rMrp2 structural changes upon drug binding. a** Displacement of the R-domain upon drug binding results in movement of both the TMDs and NBDs; both the cytosolic half-end of the TMDs and the NBDs display an 8 Å closure upon drug binding whereas no significant conformational changes are observed towards the drug binding site. **b** The two drug binding sites partly overlap with the R-domain. The probenecid bound (orange sticks) rMrp2 is shown as pink cartoons whereas the autoinhibited rMrp2 as blue cartoons. **c** Drug binding sites 1 and 2 overlap to a large extent with the substrate binding site of bMrp1. bMrp1 (grey cartoon) bound with LTC4 (green sticks) has been superimposed onto rMrp2 (pink) bound with probenecid (orange sticks). Drug binding site 2 is below the substrate binding site. The CHS molecule (grey sticks) fills the hydrophobic space of the H-pocket.

drugs;selectivity is probably due to small differences in the amino acids ligning the substrate and drug binding sites.

## Dubin–Johnson syndrome

Understanding the pathophysiology of Dubin–Johnson syndrome can shed light on the mechanism of action of MRP2. Several mutations (nonsense, missense, deletion, splice site) have been identified that result in either dysfunctional MRP2 or absence of MRP2 expression in the apical membrane[3,4,6,33]. We mapped the mutations that are associated with Dubin–Johnson syndrome onto the rMrp2 structure (Fig. 7) and interestingly none are found within the likely substrate binding site of the TMD cavity or R-domain suggesting that the observed dysfunction is not due to loss of bilirubin glucuronide recognition or regulation by phosphorylation. Most of the mutations associated with

Dubin–Johnson syndrome are located in the intracellular loops of the TMDs and at the interface of the ICL4 and NBD1, and ICL2 and NBD2, respectively, suggesting uncoupling of the conformational changes associated with signal transduction between the transport substrate binding site(s) and the ATP catalytyic cycle (Fig. 7); the interface between the two coupling helices and NBDs is critical to transmit the conformational changes associated with either formation of the NBD:NBD interface upon ATP binding to transition to an outward open conformation or disengagement of the NBDs to revert into an inward open conformation upon ATP hydrolysis. The most common mutations are R1150H and I1173F in MRP2 that are frequent in the Iranian-Jewish and Moroccan-Jewish populations that comprise the largest groups of Dubin–Johnson syndrome patients[34]. The effect of I1173F is due to low membrane insertion rather than reduced activity. R1150H is

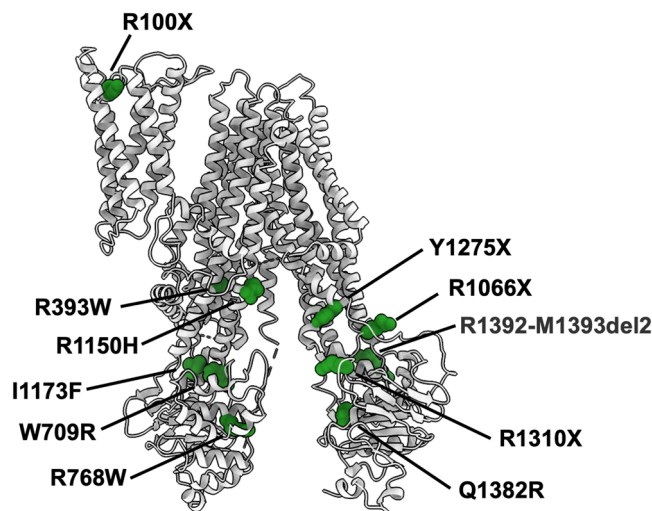

**Fig. 7 | Dubin–Johnson syndrome mutations.** Mutations that cause Dubin–Johnson syndrome have been mapped onto the rMrp2 structure; they are shown as green spheres and labelled according to the MRP2 sequence. X refers to a stop codon and del2 to a two-residue deletion. Most mutations are found on the interface of the coupling helices and the NBDs.

found at TM15 (which is likely distant from the transport substrate binding site if extrapolation from the conjugated bMrp1 LTC4 binding pocket reflects the likely binding site of bilirubin glucuronide in MRP2), suggesting that it may interfere with conformational changes along the TMD or prevent the R-domain from disengaging from the inhibited state; in the phosphorylated CFTR structure the R-domain contacts TM15.

## Discussion

Our cryo-EM structure established that the molecular architecture of rMrp2 is very similar to other members of the ABCC family, bMrp1, CFTR and Ycf1, despite their low sequence identity. A common feature between rMrp2 and the other transporters is their regulation by phosphorylation of the regulatory or R-domain. The concept of modulating the activity of transporters, both ABC- and secondary active transporters, is well established but the exact molecular mechanism of this process remains obscure. With regard to ABC transporters, our current study together with other studies provides the molecular basis to understand this regulatory process. Functional and structural studies have proposed that the R-domain interferes with NBD:NBD interaction thus reducing ATP hydrolysis and that this negative regulation can be relieved by phosphorylation[26,29,27]. Transporters with a dephosphorylated R-domain can still exhibit ATPase activity, suggesting that the inhibition by the R-domain is transient. It has been speculated that in the dephosphorylated state, the R-domain is in an equilibrium between inhibited and active-like states that is shifted towards the active state upon phosphorylation[26]; this equilibrium was proposed on the basis that dephosphorylated CFTR retained low, but not absent, probability of channel opening, that was stimulated upon phosphorylation[26]. Although the R-domain is observed deep inside the TMD of the rMRP2 structure, our functional data show that rMrp2 retains its ATPase activity adding further support for the concept of the R-domain equilibrium. Our structure provides insights on why phosphorylation of the R-domain would relieve the inhibitory state via a mechanism of R-domain movement hindrance. Our mass spectrometry data coupled to in vitro phosphorylation with human kinases and biochemical assays, ATPase and substrate transport, showed that upon full phosphorylation of the R-domain, rMrp2 displays faster ATP hydrolysis and substrate transport relative to the partially phosphorylated protein. Interestingly, the

ATPase activity of the fully phosphorylated rMrp2 could not be further stimulated in the presence of probenecid. This is in contrast to ABC transporters that do not contain an R-domain, whose ATPase activity is greatly modulated by substrates[35,36]. It is very likely that if the R-domain is partially phosphorylated, transport substrates or drugs will have to compete with the R-domain for the TMD therefore appearing to stimulate its basal ATPase activity. Phosphorylation can only happen after the R-domain has transiently moved out of the TMD or the NBD:NBD interface (as proposed for CFTR). Our mass spectrometry data point to the key role of Ser922 and Ser926 in modulating the activity of rMrp2; the purified rMrp2 had a nearly absent phospho signal for Ser922 and Ser926 that was significantly increased upon in vitro treatment; in our structure we have partly resolved the R-domain and phosphorylation of Ser922 would cause steric clashes with Asn432 and Tyr433 if the R-domain were to adopt a similar conformation to the dephosphorylated state; alternatively, phosphorylation could also stabilise the R-domain outside the TMD or NBD:NBD interface in a similar manner to the phosphorylated CFTR[26,29] and Ycf1 structures[27,28]. Although Ser926 has not been resolved in our structure, it may also contribute to steric clashes as it would still be located inside the TMD. Since the R-domain in our cryo-EM structure is partially phosphorylated, Thr869, Ser874 and Thr886, it suggests that phosphorylation of these residues does not relieve autoinhibition (full dephosphorylation of rMrp2 did not alter the ATP hydrolysis kinetics either), but their role might be to stabilise the R-domain as stated above. Our functional data support this observation, as upon phosphorylation the rMrp2 displays faster ATP turnover and CDF transport activity compared to the partially phosphorylated and fully dephosphorylated proteins. The observation of the rMrp2 autoinhibited state further supports the notion of R-domain plasticity within the TMD and at the NBD:NBD interface. We show that inhibtion of cellular kinases resulted in significantly reduced CDF export in iHEPs. Our data provide a strong correlation between the in vitro rMrp2 and *in cellulo* transport activity of MRP2, as the transport activity of the dephosphorylated rMrp2 was reduced by nearly 80% and the transport activity of human MRP2 was reduced by 50%, respectively. Overall, the transport activity of MRP2 is likely to be strongly regulated by the activity of kinases and phosphatases in the canaliculi. Despite our mechanistic insights, the exact interplay of kinases and phosphatases to regulate the phosphorylation state of MRP2 remains unclear; i.e. the exact conditions and timeframe under which MRP2 will be activated or deactivated within the cellular context.

Biochemical evidence for two drug binding sites in MRP1 and MRP2 has been reported[31,32,37,38]. We observed two molecules of probenecid in our drug-bound structure of rMrp2. Extrapolating from the bMrp1 structure with bound conjugated-LTC4 these probenecid sites likely overlap to some extent with the prospective binding site for bilirubin glucuronide in MRP2. The drug and transport substrate binding sites of MRP1 and MRP2 show some sequence conservation, but also several differences that could explain their differences in drug selectivity. Probenecid binding site 2 of rMrp2 displays sequence conservation with MRP1, whereas the probenecid binding site 1 is less conserved, suggesting a possible role in the latter for drug selectivity differecens between MRP1 and MRP2. Interestingly, probenecid is capable of either displacing the unphosphorylated R-domain or preventing it from re-entering the TMD upon the transient movement of the R-domain. Our functional data show that probenecid can stimulate the ATPase activity of partially phosphorylated rMrp2 and the structure provides insights on the modulation of the rMrp2 activity by drugs in addition to phosphorylation. The displacement of the R-domain by probenecid relieves the inhibitory state by allowing the NBD1 and NBD2 to come closer, thus likely facilitating ATP binding and hydrolysis as shown by our ATPase data. We do not yet have an ATP-bound structure of rMrp2 but would anticipate that following release from the

autoinhibited state by phosphorylation of the R-domain, bilirubin glucuronide would be able to occupy the binding cavity and induce a conformational change to allow NBD1 and NBD2 to form an interface to bind and hydrolyse ATP driving transport substrate efflux and return to the apo state. Further experiments will be required to test this hypothesis.

The observed CHS molecule is very likely recruited from the micelle as there should not be any free CHS during the later stages of purification. In our cryo-EM maps, we observe several CHS molecules in the interface of TMD0 and TMDs 1 and 2 but also a tightly bound CHS molecule between the elbow helix and TMs 14, 15 and 17, that is close to the lateral opening of the TMD. There is some evidence of cholesterol modulating the transport activity of MRP2 but not transporting it in HepG2 cells[39]. A likely role for the CHS might be to either act as a conjugate molecule of drugs such as probenecid or mask the hydrophobic pocket that is not occupied by probenecid. It has been suggested that cholesterol might be acting as a 'fill-in' molecule for the ABC transporters ABCA1 and ABCB1, when they transport small substrates by filling the empty space within the binding site[40]. This concept is relatable to rMrp2 in complex with probenecid as the two molecules do not occupy the hydrophobic P-pocket; in bMrp1 the lipid tail of LTC4 occupies this site and a cholesterol molecule is not required. Another possibility is that CHS has been recruited by probenecid as part of its modulating activity[5,17]. Probenecid is an interesting molecule as it can act as a drug modulator or inhibitor and our structure provides insights on its recognition by MRP2. Considering the drug bound structure, modulation of the activity of Mrp2 by probenecid might be due to the presence of the two drug binding sites. We propose that if probenecid is acting as an inhibitor, it quite likely occupies both sites whereas as a modulator, another drug could displace the probenecid from the drug binding site 1 that is less conserved in the ABCC family. Our structural work not only addresses how drug modulators affect its function, but it also has implications in understanding drug-drug interactions. Finally, we start getting an understanding on how mutations in Dubin–Johnson syndrome result in the dysfunction of MRP2 by interfering/disrupting the TMD and NBD interface rather than loss of substrate (bilirubin glucuronide and ATP) recognition. Overall, we show a strong interplay of modulating the activity of rMrp2 by phosphorylation and drugs and start gaining insights on how drug modulators, such as probenecid, could influence drug-drug interactions.

## Methods

### Protein expression and purification

The *rAbcc2* gene was cloned into the p424GAL1 vector with a C-terminal TEV protease site upstream of a GFP-His$_8$ and the construct was transformed into the *S. cerevisiae* FGY217 strain. Expression was performed as previously described[19]; in brief wt rMrp2 or E1458Q rMrp2 expressing strains were inoculated in 10 ml of -URA media (Yeast synthetic drop-out medium supplement without Ura) containing 2% glucose and incubated overnight in an orbital shaker at 280 RPM and 30 °C. The overnight culture was diluted in 150 mL of the same media and incubated overnight under the same conditions. The second overnight culture was diluted to an OD600 of 0.12 into 1 L of -URA containing 0.1% glucose in a 2.5-L baffled shaker flask and incubated in the same conditions until the culture reached an OD600 of 0.6. Recombinant protein expression was induced by adding 2% galactose and incubated for 22 h. After expression, the cells were harvested, and the cell pellet was flash frozen in liquid nitrogen (LN$_2$) and stored at −80 °C.

The isolated membranes were diluted to 360 mL of Membrane buffer supplemented with 2% LMNG and 0.4% CHS and incubated under mild agitation for 90 min. The unsolubilised fraction was removed by ultracentrifugation at 190,000 × g for 1 h. The supernatant was loaded onto a HisTrap HP 5 mL Ni affinity column and washed with

5 column volumes (CV) wash buffer I (50 mM Tris/HCl pH 8.0, 150 mM NaCl, 2 mM MgCl$_2$, 2 mM DTT, 5% Glycerol, 0.1% LMNG, 0.02% CHS, 70 mM imidazole), followed by 5 CV of wash buffer II (50 mM Tris/HCl pH 8.0, 150 mM NaCl, 2 mM MgCl$_2$, 2 mM DTT, 5% Glycerol, 0.02% GDN) and eluted with elution buffer (50 mM tris/HCl pH 8.0, 150 mM NaCl, 2 mM MgCl$_2$, 2 mM DTT, 5% Glycerol, 0.02% GDN, 250 mM imidazole). The protein solution was concentrated using a 100 kDa concentrator and back diluted into the same elution buffer but without imidazole to drop the imidazole concentration belove 70 mM. TEV protease was added in a 1:1 molar ratio and incubated under mild agitation overnight. The protein was loaded onto a HisTrap HP 5 mL Ni affinity column and the flow trough was collected and the column was washed with 5 CV wash buffer II. All the fractions containing rMrp2 were pulled together and applied onto a Superose 6 gel filtration column (GE Healthcare) equilibrated with SEC buffer (50 mM Tris/HCl pH 8.0, 150 mM KCl, 2 mM MgCl$_2$, 2 mM DTT, 0.02% GDN). Peak fractions were analysed by SDS-PAGE, concentrated to 5 mg/ml and were either used immediately to prepare cryo-EM grids or flash frozen in LN$_2$ for assays.

### Protein phosphorylation and dephosphorylation

Purified rMrp2$_{R-*}$ was either treated with λ-protein phosphatase or HEK293 cell extracts. To produce rMrp2$_{R-de}$, rMrp2$_{R-*}$ was mixed in a 1:1 ratio with recombinant His-tagged λ-protein phosphatase and incubated for 1 h under rotary stirring at 4 °C. To remove the λ-protein phosphatase, NiNTA beads were added to the sample and further incubated for 30 min. The sample was spun down to remove the beads with bound λ-protein phosphatase and the supernatant containing the rMrp2$_{R-de}$ was used for further experiments. To generate the rMrp2$_{R-pho}$, a HEK293 cell extract was prepared as previously described[41]; in brief, HEK293 cells were pelleted and washed once in PBS, then resuspended at 30 ×10$^6$ cell/mL in HEK293 lysis buffer (50 mM tris/HCl pH 7.5, 10 mM MgCl2, 1 mM EDTA, EDTA-free protease inhibitor tablet, PhosSTOP tablet, 1 mM PMSF) and lysed using a tissue homogenizer. The cell lysate was centrifuged at 21,000 × g to pellet cell debris and the supernatant was aliquoted in 1 mL and flash frozen in LN$_2$. All the steps were performed at 4 °C or on ice. The HEK293 cell extract was added during protein purification; after protein elution with elution buffer (50 mM tris/HCl pH 8.0, 150 mM NaCl, 2 mM MgCl2, 2 mM DTT, 5% Glycerol, 0.02% GDN, 250 mM imidazole) the protein was exchanged into HEK293 lysis buffer supplemented with 0.02% GDN. 1 mL of HEK293 cell extract and 5 mM ATPwere added to the protein and it was incubated for 30 min at 30 °C. After incubation, the mixture was re-loaded on the HisTrap column, washed with wash buffer II and eluted with elution buffer. The protein was further purified as in Protein expression and purification section.

### Mass spectrometry analysis

Protein solutions were adjusted to 100 mM triethylammonium bicarbonate (TEAB), reduced with 5 mM tris-2-carboxyethyl phosphine (TCEP), alkylated with 10 mM iodoacetamide (IAA) and digested overnight with trypsin at a final concentration of 50 ng/µL (Pierce). Peptides were SpeedVac dried and subjected to clean-up with C18 Pierce spin tips followed by phosphopeptide enrichment using the High-Select Fe-NTA Phosphopeptide Enrichment Kit (Thermo) according to manufacturer's instructions. Both the eluents (phosphopeptides) and the flowthroughs were collected for LC-MS analysis. LC-MS analysis was performed on an UltiMate 3000 UHPLC system coupled with the Orbitrap Ascend Mass Spectrometer (Thermo Scientific). Peptides were loaded onto the Acclaim PepMap 100, 100 µm × 2 cm C18, 5 µm, trapping column at flow rate 10 µL/min and analysed with a nanoEase MZ PST BEH130 1.7 µ 75 µm × 250 mm C18 capillary column. Mobile phase A was 0.1% formic acid and mobile phase B was 80% acetonitrile, 0.1% formic acid. The separation method was as follows: for 80 min gradient 5%−35% B, for 5 min

 

up to 95% B, for 5 min isocratic at 95% B, re-equilibration to 5% B in 5 min, for 5 min isocratic at 5% B, at flow rate 300 nL/min. MS scans were acquired in the range of 400–1600 m/z at mass resolution of 120 K, AGC $4 \times 10^5$ and max IT 251 ms. Precursors were selected with the top speed mode in 3 sec cycles and isolated for HCD fragmentation with quadrupole isolation width 0.9 Th. Collision energy was 35% with AGC $1.25 \times 10^5$, max IT 200 ms and orbitrap detection at 45 K resolution. Targeted precursors were dynamically excluded for 30 s with 10 ppm mass tolerance. The mass spectra were analysed in Proteome Discoverer 3.0 (Thermo Scientific) with the SequestHT and Comet search engines for peptide identification and quantification. The precursor and fragment ion mass tolerances were set at 20 ppm and 0.02 Da respectively. Spectra were searched for fully tryptic peptides with maximum 2 missed cleavages. Carbamidomethyl at C was selected as static modification and oxidation of M, deamidation of N/Q and phosphorylation of S/T/Y were selected as dynamic modifications. Spectra were searched against UniProt Rattus norvegicus reviewed protein entries, peptide confidence was estimated with the Percolator node and peptides were filtered at q-value < 0.01 based on target-decoy database search. Phosphorylation localisation probabilities were estimated with the IMP-ptmRS node and peptide quantification was performed with the Minora Feature Detector and Precursor Ions Quantifier nodes[42].

The ATPase activity of rMrp2 in different phosphorylation states was measured using the Enzcheck coupled assay kit (Thermofisher). rMrp2 and its phosphorylation variants were reconstituted into destabilised liposomes consisting of bovine liver total lipid extract (Avanti Polar Lipids). Lipids were rehydrated in a buffer consisting of 20 mM Tris pH 7.5, 150 mM NaCl at a concentration of 1 mg/ml. The opaque lipid stock was bath sonicated for 10–15 min until the solution turned clear. The small unilamellar vesicles were destabilised by the addition of 0.2% GDN. rMrp2 was added to a final concentration of 0.1 mg/ml in the lipid mixture to give a molar ratio of rMrp2 to lipid of 1:900 (mol/mol). The ATPase activity of rMrp2 in destabilised liposomes was measured in the presence of 5 mM ATP and 5 mM MgCl$_2$ at room temperature. For ligand induced ATPase activity assays, probenecid and methotrexate were dissolved in 100% DMSO to make a 100 mM stock solution and were added in the assay to the final concentrations indicated in the figure legends. The reactions were measured using a Labtech CLARIOstar Plus Microplate Reader. All measurements were performed in triplicate. Analysis and plotting of the data were performed in GraFit 5.0.13. The data were fitted in a 1$^{st}$ order rate equation $A_t = A_\infty(1-e^{-kt})$ and ATP hydrolysis rate was calculated as product of rate constant and Limit. The statistical significance of experimental data was assessed by one way ANOVA followed by Tukey test for $p < 0.025$, $p < 0.01$ and $p < 0.001$ as specified in the figure legends.

### Transport of CDF in proteoliposomes

rMrp2 at different phosphorylation states was reconstituted by detergent removal in a batch-wise procedure; in brief, mixed micelles of detergent, protein and phospholipids were incubated with 0.5 g Amberlite XAD-4 resin under rotatory stirring (1,200 r.p.m.) at 23 °C for 40 min as previously described[43]. The composition of the reconstitution was: 15 μL of rMrp2 (75 μg protein) (to avoid saturation of the fluorescent signal in transport assay, for fully phosphorylated protein only 5 μL i.e. 25 μg proteins were reconstituted), 100 μL of a mixture composed by 60 μL of 10% (w/v) egg yolk phospholipids, containing 7.5% cholesterol (w/w), in the form of sonicated liposomes and 40 μL of 10% C$_{12}$E$_8$, 70 μL PBS 10X (1× final concentration) in a final volume of 700 μL. The intraliposomal CDF (5(6)-Carboxy-2′,7′-dichlorofluorescein, Sigma-Aldrich) accumulation was monitored by measuring the fluorescence emission. After reconstitution, 600 μL of proteoliposomes was passed through a Sephadex G-75 column (0.7 cm diameter ×15 cm height) pre-equilibrated with PBS. Proteoliposomes were added of 5 μM CDF (200 mM stock solution prepared in PBD

buffer) and 1 mM probenecid (100 mM stock prepared in DMSO) as specified in the figure legend. Uptake experiments were started in a 150 μL proteoliposome sample by adding 4 mM ATP together with 4 mM MgCl$_2$ (or differently as specified in the figure legend), at 25 °C. The transport reaction was stopped after 30 min by passing the mixture through a Sephadex G-75 column (0.6 cm diameter ×8 cm height) equilibrated with PBS; samples were then, eluted with 1 mL PBS. 0.1% SDS was added to destabilise the proteoliposomes. Fluorometric measurement was performed using the fluorescence spectrometer LS55 from Perkin Elmer. The fluorescence was measured following time drive acquisition protocol with λ excitation = 504 nm and λ emission = 529 nm (slit 15/10) according to manufacturer instructions of CDF. Calibration of the fluorescence changes vs CDF concentration has been performed by measuring the fluorescence of known amounts of CDF (from 0 to 6.5 pmoles in 1 mL) obtaining a linear correlation. The calibration line was used to calculate the pmoles of CDF taken up in rMrp2-harbouring proteoliposomes. A calibration was performed at the end of each experiment. A liposome control sample i.e. liposomes without reconstituted protein was prepared to quantify the counts given from unspecific lipids fluorescence emission, controls were subtracted to each sample. The statistical significance of experimental data was assessed by one way ANOVA followed by Tukey test for $p < 0.025$ as specified in the figure legends.

### Generation of iHEPs and MRP2 assay

Human iPSCs (CGT-RCiB10, Cell and Gene Therapy Catapult) were differentiated into iHEPs essentially as described[44], except for the Matrigel (Corning) sandwich and addition of forskolin (5 μM; BioGems) from day 17. On day 19 hepatocyte growth factor was removed and taurine was added (58.4 mg/L; Sigma) and the cells are placed in normoxia. On day 25 the iHEPs were treated +/−20 nM staurosporine (Generon) for 2 h followed by addition of the MRP2 transport substrate precursor CDFDA (10 μM; Sigma-Aldrich) and NucBlue Live Cell ReadyProbes Reagent (Thermo Fisher Scientific) and incubated for a further 15 min at 37 °C. CDFDA is cleaved by intracellular esterases to yield fluorescent 5-(and-6)-carboxy-2′,7′−dichlorofluorescein (CDF; green) which is secreted into the bile canaliculi by MRP2. Cell nuclei and accumulation of CDF (green) in bile canaliculi were captured in images taken on an LSM 880 confocal microscope (ZEISS). Images were analysed using Fiji[45]. Images were first processed using global thresholding to filter out noise. Then, size-exclusion filtering was applied to remove particles <5 μm$^2$ to select biologically relevant signals to determine integrated density of CDF staining in the bile canaliculi. The statistical significance of experimental data was assessed by Student's test.

### MRP2 staining

iHEPs were treated +/−20 nM staurosporine (Generon) for 2 h, after which the cells were fixed in 4% PFA (Sigma-Aldrich) in PBS for 15 min. The fixed cells were then blocked for 1 h in blocking buffer (0.2% Triton X-100 (Sigma-Aldrich), 3% Donkey-Serum (Sigma-Aldrich) and 1% BSA (Sigma-Aldrich) in PBS) at room temperature. Next, the cells were stained with a 1:25 dilution of human MRP2 primary antibody (Abcam; ab3373) in blocking buffer overnight at 4 °C. The following day, the cells were stained with a 1:500 dilution of AlexaFluor 555 donkey anti-mouse antibody (Invitrogen; A-31570) in PBS for 2 h at room temperature, followed by a 5-min incubation with NucBlue Fixed Cell ReadyProbes Reagent (DAPI; Thermo Fisher Scientific) at room temperature. Cell nuclei (blue) and MRP2 expression at the canalicular membrane (red) were captured in images taken on an LSM 880 confocal microscope (ZEISS). Images were analysed using Fiji[45] as above to determine integrated density of MRP2 staining at the canalicular membrane, total canalicular area, average canalicular area and canalicular count. The statistical significance of experimental data was assessed by Student's test.

## Cryo-EM sample preparation and data collection

Grids were prepared with freshly purified protein, rMrp2$_{R-*}$, at 5 mg/mL. Freshly glow-discharged Quantifoil R1.2/1.3 300-mesh Cu Holey Carbon Grids were treated with 3 mM fluorinated Fos-Choline-8 (Anatrace), followed by the addition of 4 µL of rMrp2 and blotted with filter paper (blot time 3 s, blot force 3), and plunged into liquid ethane using a Vitrobot Mark IV (FEI). The same procedure was followed for the preparation of the probenecid grids; rMrp2$_{R-*}$ was incubated with 1 mM mM probenecid for 30 min prior to freezing the grids.

Cryo-EM data were collected at the eBIC (Diamond Light Source, UK) on a Titan Krios system (FEI) operated at 300 kV and images were collected using a K3 Imaging detector (Gatan). The data collection was automated, and images were collected in super-resolution mode Bin2. Data acquisition parameters are summarised in Supplementary Table 1.

## Cryo-EM Data Processing and model building

Movies were imported into cryoSPARC, corrected for beam-induced motion using patch motion correction and contrast transfer function (CTF) estimations were performed using patch CTF estimation[46]. Particles were first picked using a blob picker with minimum diameter of 100 Å and maximum diameter of 200 Å. Particles were inspected, extracted with a 440-pixel box size, and 4.3 million particles were used in an initial 2D classification with 50 2D classes (with maximum resolution of 6 Å and initial classification uncertainty factor of 2). Seven 2D classes were selected with 1.4 million particles and further reclassified into 50 2D classes with the same parameters. Eleven 2D classes were selected with 693,363 particles. A sub-batch of 174,000 particles was used to generate an ab initio model reconstruction. The initial model was used to generate 50 2D templates for a second round of particle picking using cryoSPARC's template picker, with particle diameter of 190 Å (ab initio was generate only for the nucleotide-free dataset and the generated templates were used to pick particles in both nucleotide-free and probenecid dataset). Particles were inspected and extracted resulting in a final of 3.8 and 4.9 million particles for rMrp2 nucleotide-free and drug bound, respectively. These particles were downscaled by Fourier Cropping from 440 px box to 110 px box using cryoSPARC's downsample particles job, then used in 2D classification with 50 classes, maximum resolution of 6 Å, an initial classification uncertainty factor of 2, force max over poses/shifts set to true, number of online-EM iterations set to 20, and 100 batch size per class. From this, 13 classes for nucleotide-free (1.9 million particles) and 7 classes for probenecid (1 million particles) were selected and submitted to another round of 2D classification with the same parameters. Final number on 8 classes (859,290 particles) were selected for nucleotide-free and 17 classes (421,743 particles) for probenecid. An initial 3D ab initio with three classes (maximum resolution of 12 Å, initial resolution 35 Å, class similarity 0.35) resulted in two 'junk' classes and one good class for both datasets. The one good class contained 364,687 particles (42.4%) for nucleotide-free and 247,763 particles (58.7%) for probenecid of the expected size and shape. Non-uniform refinement of the good class was performed using the downscaled particles, then the generated alignments were used to run the final non uniform refinement[47] with the original unbinned particles resulted in a map with GSFSC (0.143) estimated resolution of 3.58 Å and 3.45 Å for the nucleotide-free and drug bound rMrp2, respectively. Model building and refinement were performed starting from the AlphaFold predicted model for rMrp2 (AF-Q63120-F1). The AF-Q63120-F1 model was split into 5 domains (NBD1 and NBD2, two TMDs and the TMD0), were all individually rigid-body fitted into the map using UCSF ChimeraX[48] and real-space refined in phenix[49]. The model was then subjected to iterative cycles of refinement and manual rebuilding in COOT[50]. The refinement statistics are summarised in Supplementary Table 1.

## Model validation

The structural models were validated using MolProbity (43). Cross-validation to check possibility of overfitting and predictive power of model was performed using half-map validation approach (44).

## Reporting summary

Further information on research design is available in the Nature Portfolio Reporting Summary linked to this article.

## Data availability

The cryo-EM maps have been deposited in the Electron Microscopy Data Bank (EMDB) under accession codes EMD-19431 (nucleotide-free autoinhibited rMrp2) and EMD-19433 (probenecid bound rMrp2). The structural coordinates have been deposited to the RCSB Protein Data Bank (PDB) under the accession codes 8RQ3 (nucleotide-free auto-inhibited rMrp2) and 8RQ4 (probenecid bound rMrp2). Raw data for nucleotide-free autoinhibited rMrp2 and probenecid bound rMrp2 were submitted to Electron Microscopy Public Image Archive with IDs EMPIAR-11893 and EMPIAR-11894, respectively. The mass spectrometry proteomics data have been deposited to the ProteomeXchange Consortium via the PRIDE partner repository with the dataset identifier PXD045845. Source data are provided with this paper.

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

## Acknowledgements

We would like to acknowledge Diamond for access and support of the cryo-EM facilities at the UK national electron Bio-Imaging Centre (eBIC), proposal BC25127. We would like to thank Prof Erhard Hohenester, Imperial College London, for critical reading of the manuscript. We thank Dr. Andrew Quigley, Membrane Protein Lab, for access to the CLARIOstar plate reader. T.M. is funded by a fellowship by the project NLHT- Nanoscience Laboratory for Human Technologies, POR Calabria

FESR-FSE 14/20. J.S.C. and T.I.R. are funded by a Cancer Research UK Centre grant (C309/A25144). K.B. is funded by a Medical Research Council grant (MR/N020103/1).

## Author contributions

KB designed and managed the overall project. DD provided assistance with the initial rMrp2 expression and purification. TM performed purification, ATPase assays, transport assays in proteoliposomes, cryo-EM data collection and analysis. TM, CI and KB analysed transport data. TM and KB built and refined the structures. EG, STR and KJL performed and analysed iHEP transport assays. TIR and JC performed mass spectrometry analysis. KB wrote the manuscript with help from all the authors.

## Competing interests

The authors declare no competing interests.
