## [Peer Review File · Nature Communications]

Structural basis for the modulation of MRP2 activity by phosphorylation and drugsREVIEWER COMMENTS

Reviewer #1 (Remarks to the Author):

Mazza et al., present the nucleotide free structure of rat Mrp2 in the presence and absence of probenecid. This is work on an important protein that transports a structurally diverse set of endogenous and exogenous substances. This is one of five ABCC proteins to have structures reported and follows CFTR, Abcc1, Ycf1, and very recently ABCC4. The following are my comments on the manuscript:

1) Throughout the manuscript the importance of MRP2 in clinical anti-cancer drug resistance is emphasized as a rationale for determining its structure. However, papers cited in support of this do not correlate MRP2 levels with clinical outcomes and really the overall evidence for MRP2 involvement in drug resistance (clinically) is very limited. There is no doubt that MRP2 is a really important protein to understand but this reviewer thinks the emphasis should be on its important role in ADME and detoxification. Alternatively, better references should be included. For example, line 66-67 Introduction “Chemotherapy resistance in HCC is due to the

overexpression of MDR transporters including ABCC2 (MRP2) and ABCC3 (MRP3)” Paper cited (Nies et al., 2001) does not correlate MRP2 levels with clinical resistance. All evidence suggests that MRP2 levels go up in in vitro models.

2) Characterization of MRP2 phosphorylation has all been completed using indirect methods and not well characterized as the authors imply. The statement “MRP2 and other members of the ABCC family contain an R-domain that regulates their transport activity upon phosphorylation by specific kinases (CKII, PKC and PLK kinases).” needs to be better qualified. Nobody has directly tested the influence of phosphorylation on MRP2 activity via phospho- or dephospho-mimicking mutations followed by transport assays. Alterations in MRP2 function with pharmacological inhibitors of kinases more likely imply changes in localization not changes in function. In addition, the phosphositeplus database has very few hits for MRP2 phosphorylation using high throughput screens and no hits for low throughput screens. This paper is the first (to this reviewer’s knowledge) that has identified rat (or any other) Mrp2 phosphorylation sites directly.

3) The functional assay employed to determine the influence of phosphorylation on MRP2 function is inadequate because it is indirect. Phospho- and dephospho-mimicking mutations should be made and tested. Is staurosporine altering localization of MRP2 or altering phosphorylation resulting in reduced function? Furthermore, in the methods it reads as though staurosporine is not removed from the culture media when CDFDA is added. If staurosporine is remaining in the cultures there is a possibility that it is acting as a direct inhibitor. The ability of staurosporine to inhibit MRP2 transport should be evaluated.

4) The “R-domain” being found deep within the TMDs is unique for the ABCC proteins known so far. How would phosphorylation happen if it’s so embedded? Why is this location so different from Ycf1, CFTR, and the alpha-fold models? While Ycf1 and CFTR are not involved in anti-cancer drug transport Ycf1 and

Mrp2 have common substrates (e.g., Cd PMID: 21266201 and arsenic PMID: 10220408 and PMID: 10938093)

5) No nucleotide present in any of these structures. Since ATP would always be present physiologically this should at least be commented on in the discussion.

6) Lines 325-326 "Our functional data support this observation, as upon phosphorylation the rMRP2 displays faster kinetics compared to the dephosphorylated protein." What kinetics are being referred to here?

Minor comments:

1) throughout the manuscript Student's t-tests are used for multiple comparisons where ANOVA should be used.

2) Figure 1C: Labeling of TM helices, TMDs and cytosolic loops is not consistent with previous publications and this may create confusion. Disregarding TMD0 is not a good idea. Label all TM helices starting with 1 in TMD0, using TMD0, TMD1, and TMD2 and also numbering all cytosolic loops as for Johnson and Chen for bovine Mrp1.

3) Rat MRP2 should be rMrp2 throughout unless its referring to the gene and then it should be rAbcc2 (e.g., line 3 supplementary methods rMrp2 should be rAbcc2).

4) include all abbreviations used in figures in the legend. (e.g., Figure 3 legend define the different phosphorylation state abbreviations)

5) Typo line 444 ARTPase instead of ATPase

Reviewer #2 (Remarks to the Author):

This is a high quality paper. The authors utilise an array of methods in order to interpret and justify their results for better understanding of the structure and function of ABC transporters. Particularly commendable is that the structural observations are accompanied with thorough functional studies. Overall this is an excellent work and will be a good fit to Nature Communications.

Comments:

119 - L0, is it present in other homologous proteins, or proteins of the same family (MRP1, CFTR)? Add reference to justify its function

134-137 - Make this a bit clearer: “The R-domain is predicted to be partly helical from the AlphaFold2 model and it links NBD1 to the elbow 136 helix, but in the predicted model it was placed on the interface of the NBDs and outside the TMD whereas in our structure it sits deep inside the TMD”.

I understand the first time you mention here the predicted model, is the AlphaFold2 model, but the second time, is it still predicted by AlphaFold? Or is this model predicted by you in another way? Clarify.

200-201 - “The ATPase activity of ABC 201 transporters has been shown to be uncoupled in detergent”, please provide reference

180-183 - Possibly these results do not agree because the purified protein is in detergent, whereas the protein on which the functional assays were performed was in liposomes. These should be commented.

213-216 Is the protein in detergent or liposomes? Please, clarify

Figure S2 C appears twice on the figure, amend it

Figure S4 What is the coverage and the redundancy?

Other comments:

122 - TMD, I would say there are two TMDs composed of 6 helices each, it helps to better understand the topology shown in Figure 1

176-178 - Would be good to have a reference

204-219 - Could be good to justify why you expressed in yeast since it has not the post-translational modifications needed and then you use kinases from mammalian cells. It raises the question why not express in mammalian cells. Are there benefits of doing it this way and if yes which ones?

Reviewer #3 (Remarks to the Author):

Reviewer comments

As one of the multidrug ABC transporters, MRP2 plays a pivotal role in the translocation of diverse substrates across the cell membrane. In the submitted manuscript, Mazza, et al. present two cryo-EM structures of rat MRP2 in two distinct states: apo- and drug-bound, together with functional analysis. However, some concerns in this manuscript have to be clarified first, which are listed below. The order of my points is not ranked according to importance but rather to appearance in the main text and figures.

1. Introduction section: MRP2 has played an important role in chemotherapy resistance or chemoresistance of other cancers. However, what is the relationship between chemotherapy drugs and probenecid used for structure determination in this manuscript? Include information on probenecid and why was used for structural studies. The authors describe the apo-state and drug-bound state, both structures determined in the absence of ATP. The apo-state and nucleotide-free state of ABCC2/MRP2 is in an auto-inhibited conformation. Is the auto-inhibition linked to ATPase activity, and how does it affect substrate binding and release?
2. Line 40: there should be no “,” after “Combining”.
3. Line 105: “We expressed rMRP2 in *Sacharomyces cerevisiae*” It is “*Saccharomyces cerevisiae*”. What about the glycosylation of rMRP2 in yeast? Glycosylation is important for the transport activity of ABC transporters.
4. Line 105-109: It should be in the methods section.
5. Line 114-116: The NBD1 has a relatively low resolution from the EM density map. It is not well-defined to describe “The maps showed density with good connectivity and the presence of side chains that could be easily interpreted for model building”.
6. Line 120-121: I can't find the CHS in Fig. 1. Please refer to the figures in the main text and clarify the figure panels.

7. Line 122: Missing the reference. "The L0 linker has been shown to facilitate correct folding and trafficking of ABCC transporters."

8. Line 134-137: "The R-domain is predicted to be partly helical from the AlphaFold2 model and it links NBD1 to the elbow helix, but in the predicted model it was placed on the interface of the NBDs and outside the TMD whereas in our structure it sits deep inside the TMD." Please clarify the elbow helix and label them in the figures.

9. Line 150: ".....bMRP1 with an rmsd of 5.6 Å over 1400 Ca atoms....." It is "C?" There are so many grammar mistakes in the main text. Please find all of them and revise them.

10. Line 152-153: It is not correct to measure the distance between two NBDs if NBD1 has a relatively low resolution.

11. Line 154-158: It is a boring statement here. All the ABC transporters have these features.

12. Line 164 and line 167: Missing references.

13. What is the autoinhibited state of rMRP2? If it is true, how to explain the high basal activity?

14. The authors describe the reasoning behind the phosphorylation/de-phosphorylation experiments, and describe the contradiction between the structural and functional data but do not further elaborate on the possible causes. Has the buffer used for the function been contaminated, or ABCC2 was co-purified with endogenous ATP? AMP-PNP was previously used as a non-hydrolysable analogue of ATP and could be used as a control to the basal activity.

15. Line 174-176: "Phosphorylation of the R-domain of CFTR has been shown to stimulate its channel gating and ATPase activity, whereas dephosphorylated CFTR displays low probability of channel opening." I can't follow the statement of CFTR here. If it is true for rMRP2, why does full phosphorylated rMRP2 show less substate-stimulated ATPase activity?

16. Line 176-178: "It was proposed that the unphosphorylated R-domain is in an equilibrium between inhibited and active state that is shifted towards active state upon phosphorylation, i.e., the R-domain

can freely move in and out of the TMD." No evidence to prove the statement here. If not, the statement should be in discussion.

17. Line 180-181: "Our structure points to an autoinhibited state that ATP should not be hydrolysed as the NBDs cannot dimerise....." This is wrong and over-interpreted. The structure just shows an apo-state without substrate or ATP.

18. Line 186-190: I can't find enough evidence to prove the MS results from these figures. More experiments need to be prepared to figure out the phosphorylation of the R-domain in vitro or in vivo.

19. Lines 188-189: Ser926 is mentioned twice within the same sentence.

20. Line 193 and 194: Grammar mistake ".....prevents all the" and ".....are not compatible....."

21. Line 200-203: I can't understand why the statement of ATPase activity in detergent should be in here. It seems no relationship with the phosphorylation of rMRP2.

22. Line 204-219: Here it seems that the phosphorylation of the R-domain in rMRP2 kills the ATPase activity. This is in contrast with CFTR. I can't follow the results without strong evidence.

23. Line 220-229: This is in contrast with the ATPase activity of phosphorylated rMRP2. Perhaps it is better to add vitro transport assay to measure the rMRP2 function with phosphorylation or not.

24. Line 232-237: It belongs introduction.

25. Line 240-242: Please provide evidence of probenecid as a substrate or inhibitor.

26. Line 242-243: "Probenecid can also stimulate the basal ATPase activity of hMRP2, and we observe a similar response in our assays." Please make a new figure.

27. Line 243-244: "The cryo-EM map revealed clear density for two probenecid molecules and one CHS." Why the CHS will bind here? It is difficult to figure out CHS and cholesterol from EM density map.

28. Line 245: Grammar again. “.....have displace the R-domain.....”

29. Line 248-251: “Displacement of the R-domain from the TMD by probenecid results in a more closed overall conformation at the bottom half of the TMD and NBDs by 8 Å, respectively, whereas the top half of the TMD, where the drug density is found, displays minimal movement.” It is important to present the conformational change when bound drugs, compared to apo state.

30. Line 252-265: Mutations are necessary to figure out the interactions between drugs and key residues here.

31. Line 264-264: “.... suggesting that the TMD is not required to undergo significant conformational changes to select between substrates or drugs.” If it is true, how does MRP2/MRP1 recognize the substates?

32. Line 267-289: The paragraph has nothing to present in the manuscript. The structure predicted by AlphaFold2 can do better. It is better to show the molecular-level characterization of these mutations.

33. Lines 304-305: the statement is repetition, was previously mentioned earlier in the text.

34. Line 315: is this statement in contradiction with line 209 regarding the 48% increase in phosphorylation, leading to a nearly 100% phosphorylated protein, but the purified protein had an almost absent phosphor signal? (Line 209: 48% enhancement of rMRP2 phosphorylation, mass spectrometry showed that the protein is now fully phosphorylated.). The phosphorylation signal is nearly absent, but the R-domain is partially phosphorylated – is this a contradiction?

35. Line 335: new information included on how chemotherapy drugs modulate the expression of kinases. This was not mentioned previously in the text. Chemotherapy drugs were presented as substrates for MRP2 – see paragraph 2 of the introduction.

36. Lines 367-369: missing references on probenecid acting as a modulator or inhibitor.

37. Detergent-purified rMRP2 has a molecular weight of ~150 kDa in Fig. 1a, however, it shows one or two bands of around 180 kDa after phosphorylation in Fig. 3a.

38. In Fig. 1c, it is not suitable to label “Extracellular” and “Intracellular”, as MRP2 actually is mostly expressed in the canaliculi membrane.

39. Supplementary Fig. 1, please add the label and marker.

40. Supplementary Fig. 1d, ATPase activity is measured as a function of ATP concentration or substrate concentration. Please prepare a new figure.

41. Supplementary Fig. 2 and 5, there are no labels or scale bars of the 2D class average, please add them. The workflow needs much more information about the data processing, including micrographs, particle numbers, 2D results, initial model, 3D model, and processing details.

42. Supplementary Fig 4, trypsin cleaves specifically peptide bonds at the C-terminal side of lysine and arginine residues. It surprises me to show this figure in one research article.

43. Supplementary Fig 5, the two panels of Supplementary Fig5d has a different resolution and please change high-quality picture here.

Methods Section

1. Page 1: “After expression, the cells were harvested and the cell pellet was flash frozen in liquid nitrogen LN2) and stored at -80°C”. remove the “)”

2. It should be mentioned in the text which sample with phosphorylation or not should be used for the structural studies.

3. Page 1: “The protein solution was concentrated using a 100 KDa concentrator and back diluted into the same” kDa

4. Page 3, line 95: “The opaque lipid stock was bath sonicated for 10-15 minutes until they solution turned clear.”

5. Page 3, “The ATPase activity of rMRP2.....” activity

We would like to thank all the reviewers for their positive comments and very constructive criticism. We have tried to address all of their queries to the best of our ability and a point by point response is below:

Reviewer #1

1) Throughout the manuscript the importance of Mrp2 in clinical anti-cancer drug resistance is emphasized as a rationale for determining its structure. However, papers cited in support of this do not correlate Mrp2 levels with clinical outcomes and really the overall evidence for Mrp2 involvement in drug resistance (clinically) is very limited. There is no doubt that Mrp2 is a really important protein to understand but this reviewer thinks the emphasis should be on its important role in ADME and detoxification. Alternatively, better references should be included. For example, line 66-67 Introduction “Chemotherapy resistance in HCC is due to the overexpression of MDR transporters including ABCC2 (Mrp2) and ABCC3 (MRP3)” Paper cited (Nies et al., 2001) does not correlate Mrp2 levels with clinical resistance. All evidence suggests is that Mrp2 levels go up in *in vitro* models.

We have substantially modified the introduction which now emphasises the key role of Mrp2 in bilirubin glucuronide efflux over a possible role in multidrug resistance. Reviewer #3 had a similar comment.

2) Characterization of Mrp2 phosphorylation has all been completed using indirect methods and not well characterized as the authors imply. The statement “Mrp2 and other members of the ABCC family contain an R-domain that regulates their transport activity upon phosphorylation by specific kinases (CKII, PKC and PLK kinases).” needs to be better qualified. Nobody has directly tested the influence of phosphorylation on Mrp2 activity via phospho- or dephospho-mimicking mutations followed by transport assays. Alterations in Mrp2 function with pharmacological inhibitors of kinases more likely imply changes in localization not changes in function. In addition, the phosphositeplus database has very few hits for Mrp2 phosphorylation using high throughput screens and no hits for low throughput screens. This paper is the first (to this reviewer’s knowledge) that has identified rat (or any other) Mrp2 phosphorylation sites directly.

We agree with the Reviewer that, to our knowledge, our study is the first to identify the phosphorylation sites directly in MRP2. However, we disagree with the that the characterisation uses only indirect methods. We identified the phosphorylation sites using mass-spectrometry rather than relying on bioinformatic tools. Our ATPase data with the different forms of rMrp2 clearly show the impact of phosphorylation on its activity. We have also added new *in vitro* transport data that shows the phosphorylated rMrp2 reconstituted into liposomes exhibits significantly enhanced transport activity (see Fig 3D; an experiment requested by Reviewer #3). The data correlate well with the iHEP *in cellulo* transport data. Therefore, we believe we have submitted strong direct evidence on how phosphorylation modulates the transport activity.

The use of phospho- and dephospho-mimicking mutations is quite a complex approach as it would require us to use CRISPR to make the changes in the iHEP cells (indeed it is not possible to edit the iHEPs and this would have to be performed on the iPSCs prior to differentiation); this experiment would take well over a year to generate and characterise the mutants. Mutations in yeast would also be tricky as the yield of wt protein is very low so screening several mutations would not be straight forward and would also require substantial time. Yet our assays clearly show that non-phosphorylated Ser922 and Ser926 correlates with reduced ATPase and transport activity and that both are enhanced upon their phosphorylation. Therefore, the use of phospho- and dephospho-mimicking mutations is probably redundant in this context.

To mitigate the Reviewer’s concern about the potential for staurosporine to influence the subcellular localisation of human MRP2 we have stained for MRP2 localisation in the iHEPS and clearly show

that staurosporine does not affect its expression levels or localisation to the canaliculi and have added this new data to the supplementary information (see Fig S5).

3) The functional assay employed to determine the influence of phosphorylation on Mrp2 function is inadequate because it is indirect. Phospho- and dephospho-mimicking mutations should be made and tested. Is staurosporine altering localization of Mrp2 or altering phosphorylation resulting in reduced function? Furthermore, in the methods it reads as though staurosporine is not removed from the culture media when CDFDA is added. If staurosporine is remaining in the cultures there is a possibility that it is acting as a direct inhibitor. The ability of staurosporine to inhibit Mrp2 transport should be evaluated.

Please see reply to comment 2 as the several of the points raised here are addressed above. We agree that the presence of staurosporine in the iHEP transport assays could be interpreted as direct inhibition; this is a good point. We have therefore repeated the experiments with staurosporine added only transiently and observe no difference in the outcome. Figure 4 has been reproduced using only these data to avoid this complication.

4) The “R-domain” being found deep within the TMDs is unique for the ABCC proteins known so far. How would phosphorylation happen if it’s so embedded? Why is this location so different from Ycf1, CFTR, and the alpha-fold models? While Ycf1 and CFTR are not involved in anti-cancer drug transport Ycf1 and Mrp2 have common substrates (e.g., Cd PMID: 21266201 and arsenic PMID: 10220408 and PMID: 10938093)

Our interpretation of the data is that the R-domain is rather dynamic; the lack of clear density from the whole domain in our cryo-EM maps is evidence of this. This could explain why the rMrp2 R-domain is in a different conformation to the CFTR R-domain. This has allowed us to speculate that the R-domain is in equilibrium between inhibitory and potentiating states; both CFTR (<https://doi.org/10.1016/j.cell.2017.02.024>) and rMrp2 are still able to hydrolyse ATP when they are unphosphorylated and this can only happen if the R-domain is displaced from the TMD:TMD or NBD:NBD interface. The Ycf1 structure is fully phosphorylated and in this state would prevent the R-domain from interfacing with the TMD.

5) No nucleotide present in any of these structures. Since ATP would always be present physiologically this should at least be commented on in the discussion.

We have added a comment in the discussion.

6) Lines 325-326 “Our functional data support this observation, as upon phosphorylation the rMrp2 displays faster kinetics compared to the dephosphorylated protein.” What kinetics are being referred to here?

We have reworded to be explicit here that we refer to ATP hydrolysis and also include the new data that shows that this correlates with increased substrate transport.

Minor comments:

1) throughout the manuscript Student’s t-tests are used for multiple comparisons where ANOVA should be used.

We have changed it where appropriate to ANOVA.

2) Figure 1C: Labeling of TM helices, TMDs and cytosolic loops is not consistent with previous publications and this may create confusion. Disregarding TMD0 is not a good idea. Label all TM helices starting with 1 in TMD0, using TMD0, TMD1, and TMD2 and also numbering all cytosolic loops as for Johnson and Chen for bovine Mrp1.

We agree with the reviewer and have changed the labelling to follow the MRP1 one. Reviewer #2 had a similar comment.

3) Rat Mrp2 should be rMrp2 throughout unless its referring to the gene and then it should be rAbcc2 (e.g., line 3 supplementary methods rMrp2 should be rAbcc2).

We have corrected it.

4) include all abbreviations used in figures in the legend. (e.g., Figure 3 legend define the different phosphorylation state abbreviations)

We have defined the phosphorylation abbreviations in Fig 3 to make it easier for the reader. The abbreviations in the other figure legends are not explicitly defined as they are mentioned in the main text; the use of NBD/TMD is more standard.

5) Typo line 444 ARTPase instead of ATPase

Corrected

Reviewer #2

119 - L0, is it present in other homologous proteins, or proteins of the same family (MRP1, CFTR)? Add reference to justify its function

The L0 is found in some members of the ABCC family. We have added appropriate references for its function.

134-137 - Make this a bit clearer: “The R-domain is predicted to be partly helical from the AlphaFold2 model and it links NBD1 to the elbow 136 helix, but in the predicted model it was placed on the interface of the NBDs and outside the TMD whereas in our structure it sits deep inside the TMD”.

I understand the first time you mention here the predicted model, is the AlphaFold2 model, but the second time, is it still predicted by AlphaFold? Or is this model predicted by you in another way? Clarify.

We have changed the sentence to clarify. Both refer to the AlphaFold2 model. We have made it clearer that we refer to the AlphaFold2 model in both instances.

200-201 - “The ATPase activity of ABC 201 transporters has been shown to be uncoupled in detergent”, please provide reference

Reviewer #3 had a similar comment and we have omitted all the detergent-based data as they are not adding anything to the manuscript.

Figure S2 C appears twice on the figure, amend it

We have corrected it.

Figure S4 What is the coverage and the redundancy?

Coverage is around ~76%, and we have included this number to the figure legend. We are not sure what redundancy the Reviewer refers to.

Other comments:

122 - TMD, I would say there are two TMDs composed of 6 helices each, it helps to better understand the topology shown in Figure 1

We agree and we have relabelled the figure and throughout the text; Reviewer #1 had a similar comment.

176-178 - Would be good to have a reference

A reference has been added.

204-219 - Could be good to justify why you expressed in yeast since it has not the post-translational modifications needed and then you use kinases from mammalian cells. It raises the question why not express in mammalian cells. Are there benefits of doing it this way and if yes which ones?

The Reviewer raises an interesting point. The reason we used yeast was as part of our efforts to find stable homologues for structural studies. We assumed that the phosphorylation of Mrp2 will be undertaken by the yeast endogenous kinases. Although yeast could phosphorylate Mrp2 it was partial as we have shown. A mammalian system would also be unlikely to fully phosphorylate the protein sample due to the endogenous phosphatases. Using the mammalian cell extract will most likely ensure that the relevant kinases are present to give us full control of the conditions to manipulate the transporter.

Reviewer #3

1. Introduction section: Mrp2 has played an important role in chemotherapy resistance or chemoresistance of other cancers. However, what is the relationship between chemotherapy drugs and probenecid used for structure determination in this manuscript? Include information on probenecid and why was used for structural studies. The authors describe the apo-state and drug-bound state, both structures determined in the absence of ATP. The apo-state and nucleotide-free state of ABCC2/Mrp2 is in an auto-inhibited conformation. Is the auto-inhibition linked to ATPase activity, and how does it affect substrate binding and release?

We have substantially modified the introduction which now includes reference to probenecid, its clinical activity and its off-target inhibition of Mrp2. Reviewer #1 had a similar comment.

In regard to the comment about the autoinhibited state, we refer to the structure but also linked to the reduced ATPase activity. Throughout the manuscript we link our work to the CFTR proposition that the unphosphorylated R-domain is in an equilibrium between active and inactive states, therefore our structure has captured the R-domain conformation when the transporter is inhibited. Both dephosphorylated rMrp2 and dephosphorylated CFTR display reduced ATPase activity that supports this interpretation.

2. Line 40: there should be no “,” after “Combining”.

Corrected

3. Line 105: “We expressed rMrp2 in Sacharomyces cerevisiae” It is “Saccharomyces cerevisiae”. What about the glycosylation of rMrp2 in yeast? Glycosylation is important for the transport activity of ABC transporters.

In our mass spectrometry data, we can see glycans on the N-linked glycosylation sites. Therefore, yeast can glycosylate rMrp2.

4. Line 105-109: It should be in the methods section.

We think it is important to make a brief statement on the purification of the protein and show a figure panel as it is important for a lot of our experimental conclusions.

5. Line 114-116: The NBD1 has a relatively low resolution from the EM density map. It is not well-defined to describe “The maps showed density with good connectivity and the presence of side chains that could be easily interpreted for model building”.

While the manuscript was under review, we collected more data for the apo protein and we obtained a 3.21 Å map (local resolution is much higher for TMDs1 and 2 and TMD0). The resolution and density for the NBDs is much higher. We have updated all the figures, measurements, and models with the 3.21 Å data.

6. Line 120-121: I can’t find the CHS in Fig. 1. Please refer to the figures in the main text and clarify the figure panels.

CHS is shown as pink density in panel D. We comment in both the text and figure legend.

7. Line 122: Missing the reference. “The L0 linker has been shown to facilitate correct folding and trafficking of ABCC transporters.”

References have been added

8. Line 134-137: “The R-domain is predicted to be partly helical from the AlphaFold2 model and it links NBD1 to the elbow helix, but in the predicted model it was placed on the interface of the NBDs and outside the TMD whereas in our structure it sits deep inside the TMD.” Please clarify the elbow helix and label them in the figures.

The elbow helix is found in all ABC transporters at the amine-terminal of TM12 and it sits perpendicular to the TMD. We have clarified in the text but we did not label in the figure for better clarity.

9. Line 150: “.....bMRP1 with an rmsd of 5.6 Å over 1400 Ca atoms.....” It is “Cα”. There are so many grammar mistakes in the main text. Please find all of them and revise them.

Corrected. We have gone checked the manuscript throughout and made all relevant changes.

10. Line 152-153: It is not correct to measure the distance between two NBDs if NBD1 has a relatively low resolution.

See reply to point 5 please. We remeasured the distances in the higher resolution maps and there is no difference from the lower resolution structure.

11. Line 154-158: It is a boring statement here. All the ABC transporters have these features.

Agree and removed

12. Line 164 and line 167: Missing references.

References have been added

13. What is the autoinhibited state of rMrp2? If it is true, how to explain the high basal activity?

We have trapped a stable conformation along the transport cycle that is consistent with low protein activity. Yes, the sample is not fully inactive but it explains the low activity and we propose a model of equilibrium between active to inactive states. So, we consider the use of the autoinhibited term has merit. It is also consistent with our new transport data in proteoliposomes as shown in Figure 3D.

14. The authors describe the reasoning behind the phosphorylation/de-phosphorylation experiments, and describe the contradiction between the structural and functional data but do not further elaborate on the possible causes. Has the buffer used for the function been contaminated, or ABCC2 was co-purified with endogenous ATP? AMP-PNP was previously used as a non-hydrolysable analogue of ATP and could be used as a control to the basal activity.

We have deleted the statement as it was not clear. The protein displays ATPase activity that we link to the proposed model of equilibrium of the R-domain between active and inactive states. In our original Fig 1, we had included the E1458Q ATPase deficient mutant that shows nearly no activity. It is unlikely we have contamination from ATP as the protein has been extensively purified. In our cryo-EM 2d class averages all particles in the autoinhibited state and we do not observe any density for ATP. All the assays and structural work were undertaken with the exact same protein samples.

15. Line 174-176: “Phosphorylation of the R-domain of CFTR has been shown to stimulate its channel gating and ATPase activity, whereas dephosphorylated CFTR displays low probability of channel opening.” I can’t follow the statement of CFTR here. If it is true for rMrp2, why does full phosphorylated rMrp2 show less substrate-stimulated ATPase activity?

The increase in ATPase activity of the fully phosphorylated rMrp2 is evident in the basal ATPase activity which correlates with the new new transport data in proteoliposomes (see reply to comment 18) that show an enhanced rMrp2 transport activity following phosphorylation. The new data display similar behaviour to CFTR, that phosphorylation enhances activity.

16. Line 176-178: “It was proposed that the unphosphorylated R-domain is in an equilibrium between inhibited and active state that is shifted towards active state upon phosphorylation, i.e., the R-domain can freely move in and out of the TMD.” No evidence to prove the statement here. If not, the statement should be in discussion.

Agree and moved to discussion.

17. Line 180-181: “Our structure points to an autoinhibited state that ATP should not be hydrolysed as the NBDs cannot dimerise.....” This is wrong and over-interpreted. The structure just shows an apo-state without substrate or ATP.

We would like to disagree with the Reviewer here. In the autoinhibited conformation resolved, the drug binding sites are obscured and movement of the NBDs would be impossible without prior displacement of the R-domain. We have reworded the section to make it clearer that we are interpreting the implications of the structure for progress through the transport cycle here.

18. Line 186-190: I can’t find enough evidence to prove the MS results from these figures. More experiments need to be prepared to figure out the phosphorylation of the R-domain in vitro or in vivo.

The raw experimental MS data can be accessed in the PRIDE server. Log in details were provided in p11 of the manuscript. We had also submitted the curated MS data in an excel sheet in the online submission system.

We agree with the Reviewer, and we reconstituted rMrp2 in proteoliposomes and measured its transport activity for the compound CDF (similar to iHEPs). We clearly show that the fully phosphorylated protein exhibits a much higher transport activity relative to the partial and fully

dephosphorylated protein. The liposome transport data are in full agreement with the iHEP data. We also propose an explanation the observation that the fully phosphorylated rMrp2 cannot have its ATPase activity further stimulated upon phosphorylation.

19. Lines 188-189: Ser926 is mentioned twice within the same sentence.

Corrected

20. Line 193 and 194: Grammar mistake “.....prevents all the” and “.....are not compatible.....”

Corrected

21. Line 200-203: I can't understand why the statement of ATPase activity in detergent should be in here. It seems no relationship with the phosphorylation of rMrp2.

We agree and have removed.

22. Line 204-219: Here it seems that the phosphorylation of the R-domain in rMrp2 kills the ATPase activity. This is in contrast with CFTR. I can't follow the results without strong evidence.

To the contrary, there is enhancement of the basal ATPase activity by 70% relative to the partial or fully dephosphorylated rMrp2. Our data, align with the CFTR published data.

23. Line 220-229: This is in contrast with the ATPase activity of phosphorated rMrp2. Perhaps it is better to add vitro transport assay to measure the rMrp2 function with phosphorylation or not.

As above, we see a great enhancement of the basal ATPase but no further stimulation by drugs. We agree with the reviewer for the need of transport assays, and we reconstituted rMrp2 in liposomes and we show that upon phosphorylation there is 6-fold enhancement in the transport activity of rMrp2 compared to partial or fully dephosphorylated protein (see Fig 3D). We have discussed further in the text the relationship of transport activity and that there is no further drug-induced ATPase activity of fully phosphorylated rMrp2.

24. Line 232-237: It belongs introduction.

Agreed and moved

25. Line 240-242: Please provide evidence of probenecid as a substrate or inhibitor.

In our new proteoliposomes transport data, we show that probenecid is acting as an inhibitor, most likely competitive, for CDF.

26. Line 242-243: “Probenecid can also stimulate the basal ATPase activity of hMrp2, and we observe a similar response in our assays.” Please make a new figure.

We have not measured the effect of probenecid on human MRP2, only rMrp2, but we observed a similar stimulation as previously reported for the human MRP2. We have added the appropriate reference.

27. Line 243-244: “The cryo-EM map revealed clear density for two probenecid molecules and one CHS.” Why the CHS will bind here? It is difficult to figure out CHS and cholesterol from EM density map.

As the protein goes through extensive purification in the presence of CHS, it is unlikely that there is carry over of ergosterol from yeast (rather than cholesterol). In addition, at lower map counter levels, additional density for the hemisuccinate allowed us to identify the molecule as CHS. We refer to the cholesterol fill in model previously proposed to explain the observation that cholesterol is often resolved in complex with ABC transporters.

28. Line 245: Grammar again. “.....have displace the R-domain.....”

Corrected

29. Line 248-251: “Displacement of the R-domain from the TMD by probenecid results in a more closed overall conformation at the bottom half of the TMD and NBDs by 8 Å, respectively, whereas the top half of the TMD, where the drug density is found, displays minimal movement.” It is important to present the conformational change when bound drugs, compared to apo state.

The original Fig 6A showed the differences in the TMD level; we have now added a new panel B to show the lack of substantial conformational differences close to the binding site.

30. Line 252-265: Mutations are necessary to figure out the interactions between drugs and key residues here.

Our study is focusing on the modulation of the activity of Mrp2 by phosphorylation and the drug modulator probenecid, and although mutagenesis studies with drugs is an excellent suggestion, we believe that they should be on a different study.

31. Line 264-264: “.... suggesting that the TMD is not required to undergo significant conformational changes to select between substrates or drugs.” If it is true, how does Mrp2/MRP1 recognize the substates?

Since Mrp2 and other members of the ABCC family are promiscuous transporters, the residues within the H pocket, P-pocket and two drug binding sites can recognise the different substrates. Figure 6B and 6C clearly show that there is minimal conformational difference between the apo rMrp2, probenecid-bound rMrp2 and the LTC4 bound bMrp1. Since our work is discussing modulation of activity, we prefer to refrain from detailed discussing multi-drug recognition,.

32. Line 267-289: The paragraph has nothing to present in the manuscript. The structure predicted by AlphaFold2 can do better. It is better to show the molecular-level characterization of these mutations.

We are not sure how an AlphaFold2 model will provide more insights than mapping the mutations on an experimentally determined structure. We include an explanation of the molecular basis of some of these mutations based on our structure.

33. Lines 304-305: the statement is repetition, was previously mentioned earlier in the text.

Removed

34. Line 315: is this statement in contradiction with line 209 regarding the 48% increase in phosphorylation, leading to a nearly 100% phosphorylated protein, but the purified protein had an almost absent phosphor signal? (Line 209: 48% enhancement of rMrp2 phosphorylation, mass spectrometry showed that the protein is now fully phosphorylated.). The phosphorylation signal is nearly absent, but the R-domain is partially phosphorylated – is this a contradiction?

We agree with the Reviewer that this statement is confusing. Line 315 refers to the absence of phosphor signal for Ser922 and Ser926 not the whole R-domain. We have edited the sentence to clarify that we mean these two serines.

35. Line 335: new information included on how chemotherapy drugs modulate the expression of kinases. This was not mentioned previously in the text. Chemotherapy drugs were presented as substrates for Mrp2 – see paragraph 2 of the introduction.

We agree that this is causing confusion. We have deleted the statement. We have added a new open question regarding the conditions under which the kinases can phosphorylate or the phosphatases dephosphorylate Mrp2.

36. Lines 367-369: missing references on probenecid acting as a modulator or inhibitor.

References have been added.

37. Detergent-purified rMrp2 has a molecular weight of ~150 kDa in Fig. 1a, however, it shows one or two bands of around 180 kDa after phosphorylation in Fig. 3a.

The double band is an artifact of SDS-PAGE. We have re-run the sample from the SEC in Fig 1a under the same condition and we also observe the same double band. We changed Fig1a to the double band to avoid the confusion. We had also mislabelled the MW on Fig 3A and it also has a band at ~150kDa.

38. In Fig. 1c, it is not suitable to label “Extracellular” and “Intracellular”, as Mrp2 actually is mostly expressed in the canalicular membrane.

Exposed to the canalicular lumen is extracellular. We have changed the labels to be more specific which membrane we refer too.

39. Supplementary Fig. 1, please add the label and marker.

Added

40. Supplementary Fig. 1d, ATPase activity is measured as a function of ATP concentration or substrate concentration. Please prepare a new figure.

We have decided to remove the ATPase data as a response to point 21.

41. Supplementary Fig. 2 and 5, there are no labels or scale bars of the 2D class average, please add them. The workflow needs much more information about the data processing, including micrographs, particle numbers, 2D results, initial model, 3D model, and processing details.

We changed the figures to provide more information.

42. Supplementary Fig 4, trypsin cleaves specifically peptide bonds at the C-terminal side of lysine and arginine residues. It surprises me to show this figure in one research article.

We believe this figure is quite important to easily convey which parts of the sequence are covered, especially in respect to the identification of the phosphorylation sites in the R-domain.

43. Supplementary Fig 5, the two panels of Supplementary Fig5d has a different resolution and please change high-quality picture here.

Changed

Methods Section

1. Page 1: “After expression, the cells were harvested and the cell pellet was flash frozen in liquid nitrogen LN2) and stored at -80°C”. remove the “)”

Corrected

2. It should be mentioned in the text which sample with phosphorylation or not should be used for the structural studies.

We altered the text to indicate that we used the partially phosphorylated protein for the structural studies.

3. Page 1: “The protein solution was concentrated using a 100 KDa concentrator and back diluted into the same” kDa

Changed to kDa

4. Page 3, line 95: “The opaque lipid stock was bath sonicated for 10-15 minutes until they solution turned clear.”

Corrected

5. Page 3, “The ATPase activity of rMrp2.....” activity

Corrected

REVIEWERS' COMMENTS

Reviewer #1 (Remarks to the Author):

The authors have appropriately addressed my previous concerns.

A few minor modifications as follows:

1) statement "Biochemical evidence for two drug binding sites in MRP1 and MRP2 has been reported" needs more references. Lots of mutagenesis and functional studies has supported at least two substrate binding sites (e.g., PMID: 11500505, previous bovine Mrp1 structures, PMID: 31268744)

2) Line 339- refer to CDF transport kinetics. Where are these data? I think "transport activity" is more appropriate than "transport kinetics".

Reviewer #2 (Remarks to the Author):

The authors have addressed all comments and this is now acceptable for publication.

Reviewer #3 (Remarks to the Author):

These authors have addressed all my concerns and I am happy with these changes. I have no further concerns. The revised manuscript is fine and can be accepted.

Reviewer #1

1) statement "Biochemical evidence for two drug binding sites in MRP1 and MRP2 has been reported" needs more references. Lots of mutagenesis and functional studies has supported at least two substrate binding sites (e.g., PMID: 11500505, previous bovine Mrp1 structures, PMID: 31268744)

We included the suggested references.

2) Line 339- refer to CDF transport kinetics. Where are these data? I think "transport activity" is more appropriate than "transport kinetics".

We changed to transport activity.